# Structural basis for distinct inflammasome complex assembly by human NLRP1 and CARD8

Qin Gong[1,2,9], Kim Robinson[3,9], Chenrui Xu[1,2], Phuong Thao Huynh[1,2,4], Kelvin Han Chung Chong[1,2], Eddie Yong Jun Tan[1,2], Jiawen Zhang[1,2], Zhao Zhi Boo[1,2], Daniel Eng Thiam Teo[5], Kenneth Lay [5], Yaming Zhang[1,2], John Soon Yew Lim[3], Wah Ing Goh[3], Graham Wright[5], Franklin L. Zhong [3,4,5✉], Bruno Reversade [5,6,7,8✉] & Bin Wu [1,2✉]

Nod-like receptor (NLR) proteins activate pyroptotic cell death and IL-1 driven inflammation by assembling and activating the inflammasome complex. Closely related sensor proteins NLRP1 and CARD8 undergo unique auto-proteolysis-dependent activation and are implicated in auto-inflammatory diseases; however, their mechanisms of activation are not understood. Here we report the structural basis of how the activating domains (FIIND^UPA-CARD) of NLRP1 and CARD8 self-oligomerize to assemble distinct inflammasome complexes. Recombinant FIIND^UPA-CARD of NLRP1 forms a two-layered filament, with an inner core of oligomerized CARD surrounded by an outer ring of FIIND^UPA. Biochemically, self-assembled NLRP1-CARD filaments are sufficient to drive ASC speck formation in cultured human cells— a process that is greatly enhanced by NLRP1-FIIND^UPA which forms oligomers in vitro. The cryo-EM structures of NLRP1-CARD and CARD8-CARD filaments, solved here at 3.7 Å, uncover unique structural features that enable NLRP1 and CARD8 to discriminate between ASC and pro-caspase-1. In summary, our findings provide structural insight into the mechanisms of activation for human NLRP1 and CARD8 and reveal how highly specific signaling can be achieved by heterotypic CARD interactions within the inflammasome complexes.

[1] School of Biological Sciences, Nanyang Technological University, Singapore 637551, Singapore. [2] NTU Institute of Structural Biology, Nanyang Technological University, Singapore 636921, Singapore. [3] Skin Research Institute (SRIS), Agency of Science Technology and Research (A*STAR), 8A Biomedical Grove, #06-06 Immunos, 138648 Singapore, Singapore. [4] Lee Kong Chian School of Medicine, Nanyang Technology University, 11 Mandalay Road, 308232 Singapore, Singapore. [5] Institute of Molecular and Cell Biology, Agency of Science Technology and Research (A*STAR), 61 Biopolis Dr, 138673 Singapore, Singapore. [6] Genome Institute of Singapore, Agency of Science Technology and Research (A*STAR), 8A Biomedical Grove, #06-06 Immunos, 138648 Singapore, Singapore. [7] Department of Paediatrics, Yong Loo Lin School of Medicine, National University of Singapore, 10 Medical Dr, 117597 Singapore, Singapore. [8] The Medical Genetics Department, School of Medicine (KUSoM), Koç University, 34010 Istanbul, Turkey. [9]These authors contributed equally: Qin Gong, Kim Robinson. ✉email: franklin.zhong@ntu.edu.sg; bruno@reversade.com; wubin@ntu.edu.sg

Nod-like receptor (NLR) proteins are cytosolic sensors and activators of the innate immune response. Many NLR proteins nucleate the "inflammasome"—a large macro-molecular complex that triggers inflammatory cell death and cytokine secretion in response to microbial or danger-associated ligands. Like many other supramolecular assemblies in the human innate immune system[1,2], the inflammasome complex relies on homotypic and heterotypic interactions between the death domains (DDs) to initiate, amplify, and propagate pyroptotic signaling[3,4]. Inflammasome sensor NLRs can be grouped into two categories based on their DDs: PYRIN domain (PYD) and caspase activation and recruitment domain (CARD)[5–7], both domains were found to form filamentous complexes to initiate the inflammasome formation. Recently, we and others have expanded the repertoire of CARD-based human inflammasome sensors to include NLRP1 and a related receptor, CARD8[8–11], both of which are implicated in human autoinflammatory diseases (Fig. 1a)[9,12–14]. NLRP1 and CARD8 both harbor a poorly defined auto-proteolytic domain known as FIIND, whose auto-cleavage is required for inflammasome activation[15,16]. The requirement of the adaptor protein ASC has also been a matter of debate for CARD-containing sensor proteins such as NLRP1 and NLRC4[17]. Very recently, human CARD8, which is absent from rodents, was found to bypass ASC altogether and activate caspase-1 directly[18].

In this study, we investigate the structural basis of how the auto-proteolysis-dependent, activating domains of human NLRP1 and CARD8 (FIIND$^{UPA}$-CARD) trigger inflammasome assembly and activation. We demonstrate that FIIND$^{UPA}$ and CARD domains play non-redundant roles in NLRP1 activation: NLRP1-CARD is sufficient to form filaments in vitro, trigger ASC speck formation and activate pyroptosis in human cells, albeit with limited efficiency. NLRP1-FIIND$^{UPA}$ does not interact with ASC directly, but dramatically lowers the threshold of NLRP1-CARD oligomerization and filament formation. Mechanistically, stand-alone FIIND$^{UPA}$ forms ring-like oligomers, whereas recombinant FIIND$^{UPA}$-CARD forms a two-layered filament, with oligomerized CARD as the inner core around which a FIIND$^{UPA}$ ring spirals. Further analysis reveals that a 9–22-amino-acid linker between FIIND and CARD contributes to the distinct helical symmetry of the NLRP1- and CARD8- FIIND$^{UPA}$-CARD two-layered filaments. By solving the cryo-EM structures NLRP1- and CARD8-CARD core filaments at 3.7 Å, we map the precise structural motifs that mediate heterotypic interactions between the CARD domains of NLRP1, CARD8, ASC, and CASP1. This analysis provides a structural explanation as to why activated NLRP1 avidly triggers ASC speck formation, whereas CARD8 directly activates pro-caspase-1. In conjunction with three complementary[19–21] preprint manuscripts published while this manuscript was in revision, our findings unveil the structural basis for NLRP1 and CARD8 inflammasome assembly and suggest that intrinsic structural properties of the CARD domain can determine the specificity of inflammasome signaling.

## Results

### NLRP1-FIIND$^{UPA}$ domain facilitates NLRP1-CARD oligomerization.
Previously, we and others have demonstrated that C-terminal auto-cleavage fragments of NLRP1 and CARD8 (NLRP1-FIIND$^{UPA}$-CARD and CARD8-FIIND$^{UPA}$-CARD) are sufficient to drive pyroptosis in cultured human cells[9,11,22] (Fig. 1a). To further map the exact role of FIIND$^{UPA}$, we utilized the widely used "ASC-speck" assay in HEK239T cells expressing GFP-tagged full-length ASC (293T-ASC-GFP). Overexpressed NLRP1-CARD is sufficient to induce ASC-GFP specks, but does so much less readily as compared with the FIIND$^{UPA}$-CARD

fragment, in agreement with previous findings[9,22]. This difference is especially notable at lower levels of protein expression (Fig. 1b, 50 ng; Fig. 1c, lane 3 and 6). At limiting protein expression levels, ~90% of NLRP1-CARD exists in the monomeric form (Fig. 1d, lane 2 and 3), with the rest as higher-molecular weight oligomers, in agreement with its low but detectable activation capacity. In contrast, FIIND$^{UPA}$-CARD, at comparable expression levels, exists mostly as high-molecular-weight oligomeric species (Fig. 1d, lane 1 vs. 2 and 3). Thus, although NLRP1-CARD is sufficient to induce ASC specks when oligomerized at elevated concentrations, the presence of FIIND$^{UPA}$ domain strongly enhances its activation by lowering the threshold of oligomerization.

Next, we examined recombinant NLRP1-CARD, FIIND$^{UPA}$-CARD, and a 1:1 molar mixture of FIIND$^{UPA}$ with CARD using negative stain-electron microscopy (EM). In agreement with the native PAGE results in 293 T cells, NLRP1-FIIND$^{UPA}$ lowered the concentration requirement for CARD oligomerization by at least 50-fold (Fig. 1e, f), but only when expressed along with CARD as a single polypeptide (Fig. 1f). NLRP1-CARD, either in isolation or incubated with FIIND$^{UPA}$ as a separate polypeptide, was unable to form ordered oligomers at concentrations <~200 μM, but did appear as long ordered oligomers at concentrations at 500 μM or above (Fig. 1e, right, Fig. 1f). These results suggest that NLRP1-CARD has a limited but detectable propensity for self-oligomerization. The FIIND$^{UPA}$ domain, akin to a "built-in" catalyst, strongly lowers the threshold for NLRP1-CARD oligomerization.

To seek direct evidence for this model, we purified the following NLRP1 fragments: FIIND$^{UPA}$-CARD, full-length FIIND-CARD (FIIND$^{fl}$-CARD), FIIND$^{UPA}$, and CARD alone (Supplementary Fig. 1). Both HEK293T cell-derived and bacteria-derived FIIND$^{UPA}$-CARD formed long heterogenous filaments ~14–15 nm in diameter (Supplementary Fig. 1a, b, e), but not FIIND$^{UPA}$ alone (Supplementary Fig. 1c), or FIIND$^{fl}$-CARD (Supplementary Fig. 1d). In agreement with its detectable oligomerization capacity in 293 T cells, recombinant NLRP1-CARD also formed filaments, similar to CARD domains found in other innate immune complexes. As expected, these filaments were visibly thinner than those formed by FIIND$^{UPA}$-CARD (~8 nm in diameter) (Supplementary Fig. 1b, e).

We next set out to solve the structures of NLRP1-FIIND$^{UPA}$-CARD and the equivalent activating fragment of CARD8. Detailed construct optimization was conducted to determine the roles of linkers and exact domain boundaries to facilitate structure determination (Supplementary Fig. 2). Truncation analysis demonstrated that the first 11 residues of NLRP1-FIIND$^{UPA}$ (a.a. 1213-1223), 11 residues from the linker region (a.a.1360–1370), and the last eight amino acids of CARD (1466–1473) were dispensable for NLRP1-FIIND$^{UPA}$-CARD to induce ASC-GFP speck formation and oligomer formation (Supplementary Fig. 2a–c). However, these shorter constructs did not improve filament homogeneity in vitro; hence the full NLRP1-FIIND$^{UPA}$-CARD filaments (a.a. 1213–1473) were used for subsequent structural studies (Supplementary Fig. 1a, e). Under negative stain EM, multiple 2D classes with distinct rotational symmetries were present with various diameters and helical settings (Fig. 1g). In all, 3D projection of the most abundant 15 nm 2D class yielded a helical filament with 11–12 identical subunits per turn and ~5.6 nm per helical rise, which is coincidentally similar to the rise of NLRC4 full-length disc-like complexes[23]. Notably, when examined using differing contour levels, this NLRP1-FIIND$^{UPA}$-CARD filament class was found to consist of a denser inner core filament with a diameter of ~8 nm (Fig. 1h). Surrounding this inner core is a discrete outer layer that extends to 15 nm across the center decorated with distinct "blobs"

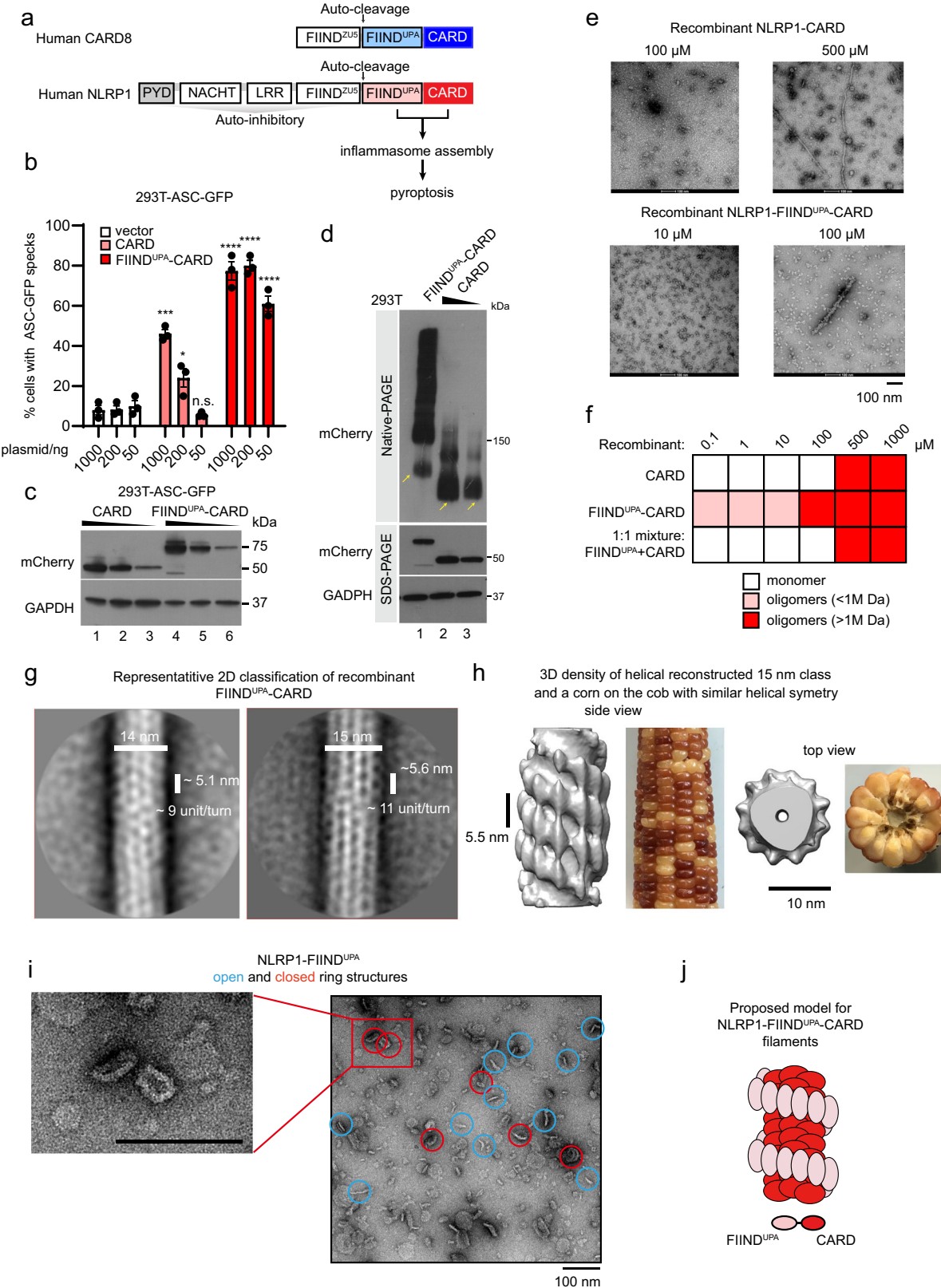

of protruding density, which resembles a corn cob (Fig. 1h). We were fascinated to identify such a "two-layer" architecture for a FIIND-containing inflammasome sensor. It is reminiscent of activated NLRC4 and NLRP6[24,25], where the nucleotide-binding domain (NBD) pre-oligomerizes to bring the CARD or PYDs into close proximity. The observed dimensions of the inner core

filament, akin to the stall of the corn cob are consistent with those of the recombinant filaments composed of CARD alone (Supplementary Fig. 1b, 8 nm in diameter, see below). The most likely explanation is that the FIIND$^{UPA}$-CARD oligomers consist of an inner helical filament of oligomerized CARDs surrounded by a FIIND$^{UPA}$ spiral facing outward (Fig. 1h). As further

**Fig. 1 Human NLRP1 inflammasome complexes. a** Domain organizations of human NLRP1, CARD8. Death domains (PYD and CARD) are highlighted in blue or red. **b** Percentage of ASC-GFP specks induced by different amounts of mCherry-tagged NLRP1-FIIND<sup>UPA</sup>-CARD (red) or mCherry-tagged NLRP1-CARD (light red) in HEK293T-ASC-GFP cells. Cells were fixed 24 hours after transfection for fluorescence microscopy. Data are presented as mean values ± SD. P value was calculated with one-way ANOVA, n = 3 biological replicates. '*/**/****' indicates P value < 0.05, 0.01, and 0.0001, correspondingly. **c** Corresponding western blot of mCherry-tagged NLRP1 constructs in Fig. 1b. **d** Blue-native PAGE blot of mCherry-tagged NLRP1-FIIND<sup>UPA</sup>-CARD and mCherry-tagged NLRP1-CARD. Yellow arrows, monomeric species. **e** Representative negative stain EM images recombinant NLRP1-FIIND<sup>UPA</sup>-CARD and NLRP1-CARD at various concentrations. **f** A summary of oligomer formation of various NLRP1 constructs at various concentrations. **g** Two of the best 2D class images with clearly distinguishable features of recombinant NLRP1-FIIND<sup>UPA</sup>-CARD filamentous complex, from negative stain EM analysis. **h** A side view and a top view of the 3D density model at estimated 20–30 Å resolution (after 3D refinement using RELION) generated based on particles extracted from the 15 nm 2D class. A similar heterogenous helical symmetry can be found in some corn cobs. **i** Negative EM images of SUMO-tagged NLRP1-FIIND<sup>UPA</sup>. The ring-complexes are ~20 nm in size. **j** Proposed model for NLRP1 FIIND<sup>UPA</sup>-CARD oligomers.

evidence for this model, recombinant SUMO-tagged NLRP1-FIIND<sup>UPA</sup> spontaneously formed ring-shaped or linear oligomers of heterogeneous stoichiometries in vitro. These closed "rings" had a minimum diameter of ~17 nm (Fig. 1i, red circles). Accommodating the SUMO tag, this is consistent with the dimensions of the outer periphery of the NLRP1-FIIND<sup>UPA</sup>-CARD filament, similar to the kernels of the corn cob (Fig. 1h–j). Similar biochemical analysis of the FIIND<sup>UPA</sup>-CARD and FIIND<sup>UPA</sup> domains from CARD8 revealed that CARD8-FIIND<sup>UPA</sup> most likely functions in a similar way as those of NLRP1. Recombinant CARD8-FIIND<sup>UPA</sup> also formed oligomers, although the oligomer architecture appears to be more heterogeneous in nature (Supplementary Fig. 3a–c). To improve the homogeneity and/or stability of the CARD8-FIIND<sup>UPA</sup>-CARD filaments, we engineered a 14-residue linker between the UPA and CARD domain (Supplementary Fig. 2d–h), which did not affect the ability of CARD8-FIIND<sup>UPA</sup>-CARD to induce CASP1-CARD oligomerization (Supplementary Fig. 2f–h). Negative stain EM images of both wild-type and 14 a.a. linker constructs were pooled together, filaments were picked and analyzed by 2D classification (Supplementary Fig. 3d). A clear two-layer filament feature was seen after enhancing the contrast of the 2D class images (Supplementary Fig. 3d, right). We then carried out 3D classification of selected 2D classes without pre-defining any helical symmetry (Supplementary Fig. 3e, f). Despite the significant heterogeneity, the overall architecture of CARD8-FIIND<sup>UPA</sup>-CARD resembles that of NLRP1-FIIND<sup>UPA</sup>-CARD (Supplementary Fig. 3f). In further support of the two-layer filament architecture, the top view of the CARD8-FIIND<sup>UPA</sup>-CARD 3D class images, after applying the helical symmetry, closely resembled that of untagged CARD8-FIIND<sup>UPA</sup>-CARD after refolding (Supplementary Fig. 3g, h). Hence, despite being less organized, CARD8-FIIND<sup>UPA</sup>-CARD also forms two-layered filaments with similar dimensions and arrangement as NLRP1-FIIND<sup>UPA</sup>-CARD (Supplementary Fig. 3h, i). These findings offer a structural explanation for the necessity for the FIIND<sup>UPA</sup> domain in aiding CARD oligomerization in both NLRP1 and CARD8. Once triggered, NLRP1-FIIND<sup>UPA</sup> and CARD8-FIIND<sup>UPA</sup> pre-oligomerizes to bring the respective CARD domains in close proximity to activate downstream inflammasome signaling.

**Cryo-EM structures of NLRP1- and CARD8-CARD filaments: the inner core of the FIIND<sup>UPA</sup>-CARD filaments**. It was recently reported that human NLRP1 and CARD8, long thought to be functional paralogues, can give rise to distinct signaling outcomes: while NLRP1 avidly triggers ASC speck formation, CARD8 can directly activate pro-caspase-1 without the need for ASC[18]. The mechanistic basis for this disparity is currently unclear. Considering our finding demonstrating that NLRP1-CARD, once oligomerized, is sufficient to activate downstream inflammasome signaling (Fig. 1b), we hypothesized that this

difference could be owing to the intrinsic structural differences between NLRP1-CARD and CARD8-CARD. To test this, we solved the structures of recombinant NLRP1-CARD and CARD8-CARD filaments using cryo-EM, both at 3.7 Å with Gold standard FSC = 0.143, without masking (Fig. 2a–i, Supplementary Fig. 4a and Supplementary Table 1). NLRP1-CARD filaments are relatively flexible and adopt multiple helical organizations in vitro, as evidenced by heterogeneous 2D and 3D classes observed during structural analysis using RELION 3.0 (Supplementary Fig. 4b)[26]. One out of four observed 3D class conformations of NLRP1-CARD filaments was determined to be suitable for high-resolution structural determination[27] (Supplementary Fig. 4c, Fig. 2b–e). Trimming of the last eight a.a. (1466–1473) helped stabilize the homogenous form of NLRP1-CARD single filament (Supplementary Fig. 1e). This conformation was subsequently verified to be a signaling-competent confirmation based on functional experiments using 293T-ASC-GFP cells (see below). Importantly, the resolved CARD filament density fits snugly into the lower resolution model of NLRP1-FIIND<sup>UPA</sup>-CARD filament based on negative stain EM, matching the shape and dimensions of the inner layer almost perfectly (Fig. 2e). Similar to NLRP1-CARD, recombinant CARD8-CARD also formed ~8 nm-wide filaments, but these filaments are more rigid and adopt a single conformation (Fig. 2f–h, and Supplementary Fig. 5a, b). The final polished densities demonstrated clear α-helix pitch and supported the unambiguous assignment of all bulky side chains (Fig. 2d, i, Supplementary Fig. 5a–d) in both filaments.

NLRP1-CARD and CARD8-CARD filaments display similar helical parameters (5.36 Å and −100.8 degree for NLRP1, and 5.409 Å and −99.16 degree for CARD8), suggesting a similar mode of assembly from the respective monomers (Fig. 2d, i). Both filaments also have clearly demarcated Type I–III interfaces that are characteristic of oligomerized DD (Fig. 2j–k, Supplementary Fig. 5c, d)[2,24,28,29]. The definition of the Type I–III interface follows the naming convention first used in describing the MyDDosome complex[2], and subsequently widely adopted in describing DD filaments, such as MAVS-CARD. When positioning the first alpha-helix of CARD domain downward, and the sixth alpha-helix upward, the monomer on top is called "a" monomer, and all three neighboring monomers are called "b" monomers. From left to right, each "b" monomers have a unique interaction interface with "a" monomer, and from left to right, named as Type I, Type II, and Type III interfaces, respectively. The corresponding surfaces on "a" monomer are Type Ia, IIa, and IIIa surfaces, and the surfaces on the three "b" monomers are type Ib, IIb, and IIIb surfaces. All surface interactions within DD filaments could be grouped into one of these types of interfaces (Fig. 2d, i). As compared with other DD complexes, the Type II interfaces in NLRP1- and CARD8-CARD filaments display stronger and more-prominent electrostatic interactions than Type I and Type III (Fig. 2j, k). This is particularly notable in

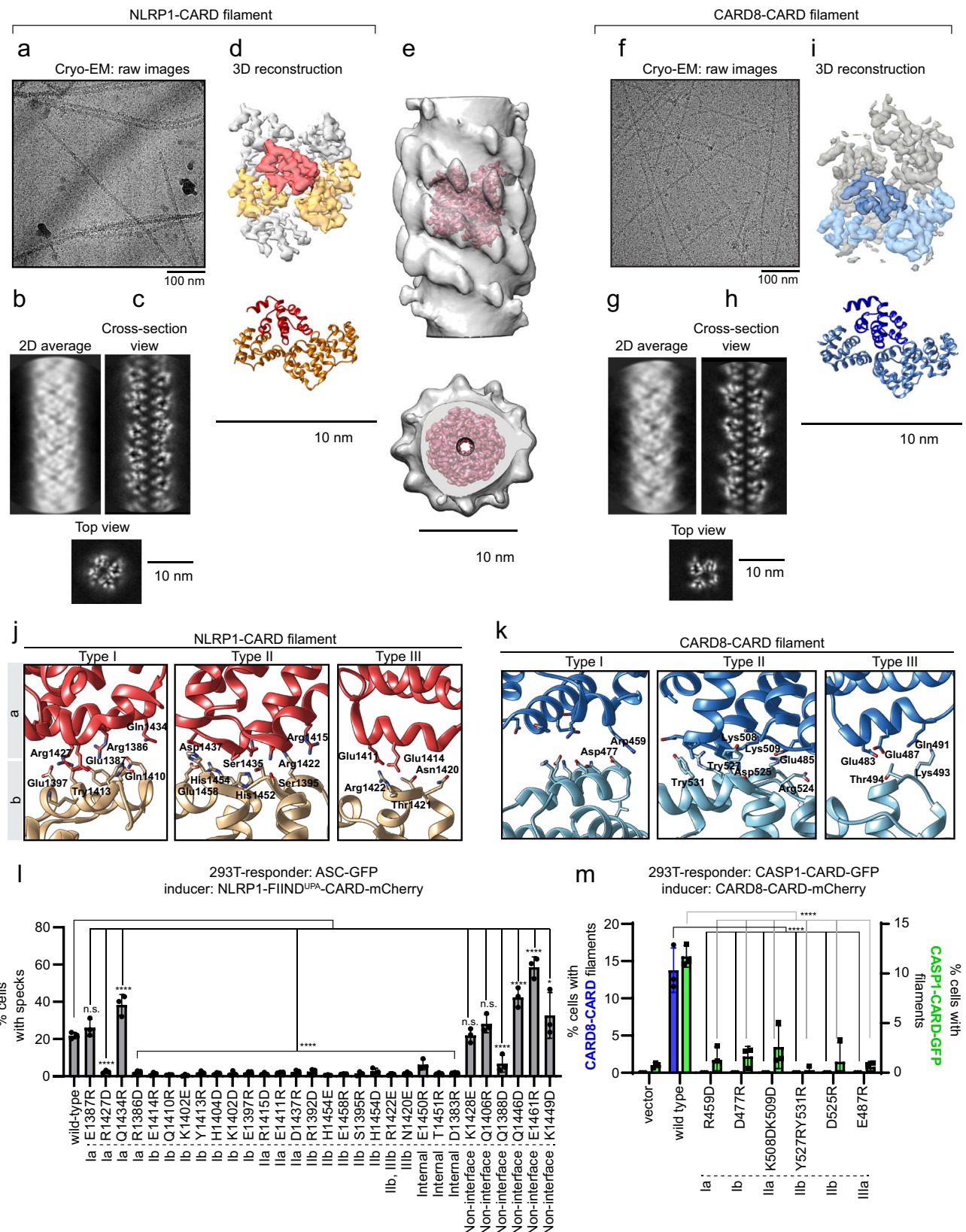

CARD8-CARD filaments, where Type Ia and Ib surfaces are separated from each other (Fig. 2j, k). To validate these structures functionally, predicted type I–III interface residues were mutated by site-directed mutagenesis. For both filaments, missense mutations of Type I–III interface residues abrogated filament assembly as well as their ability to nucleate downstream ASC-GFP or CASP1-CARD-GFP specks/filaments (Fig. 2l, m), whereas non-interface surface-exposed residues have less effect. These results further prove that CARD oligomerization is a prerequisite for NLRP1 and CARD8-driven inflammasome activation. A notable exception is the type Ia interface of NLRP1, where two of the four mutations, E1387R and Q1434R, did not affect ASC-GFP

**Fig. 2 Cryo-EM structures of NLRP1-CARD and CARD8-CARD filaments. a** Cryo-EM images of recombinant NLRP1-CARD filaments. **b** 2D average of NLRP1-CARD filaments. **c** Cross-section and top view of NLRP1-CARD filaments. **d** Helical reconstructed NLRP1-CARD complex density and molecular models. Adjacent NLRP1-CARD monomers are colored in light red ("a" monomer) and orange ("b" monomers). **e** Atomic resolution NLRP1-CARD filament density fit into the inner layer density of NLRP1-FIIND$^{UPA}$-CARD shown in Fig. 1h. **f** Cryo-EM raw images of recombinant CARD8-CARD filaments. **g** 2D average of CARD8-CARD filaments. **h** Cross-section and top view of CARD8-CARD filaments. **i** Helical reconstructed CARD8-CARD complex density and molecular models. Adjacent CARD8-CARD monomers are colored in blue ("a" monomer) and light blue ("b" monomers). **j** Type I–III interfaces in NLRP1-CARD filaments. **k** Type I–III interfaces in CARD8-CARD filaments. **l** Mutagenesis of critical NLRP1-CARD interface residues and their effects on NLRP1-FIIND$^{UPA}$-CARD-mCherry to induce ASC-GFP specks in 293 T cells. Data are presented as mean values ± SD. P value was calculated with One-way ANOVA, n = 3 biological replicates. '\*/\*\*/\*\*\*\*' indicates P value < 0.05, 0.01, and 0.0001, correspondingly. **m** Mutagenesis of critical CARD8-CARD interface residues and their effects on CARD8-CARD-mCherry filament formation (blue) and CASP1-CARD-GFP oligomerization in 293 T cells (green). Data are presented as mean values ± SD. P value was calculated with one-way ANOVA, n = 3 biological replicates. '\*\*\*\*' indicates the individual pairwise one-sided statistical analysis of wild-type and mutant has a P value < 0.0001.

speck formation in HEK293T cells despite partially reducing NLRP1-CARD filament assembly in vitro (Fig. 2l, Supplementary Fig. 4d). We postulate that the Type Ia (bottom-facing) interface of NLRP1 is predominantly involved in NLRP1-CARD self-oligomerization, rather than NLRP1-ASC interaction.

**FIIND$^{UPA}$ organization in the outer layer of FIIND$^{UPA}$-CARD filament.** Next, we carried out scanning mutagenesis to define the structural determinants for NLRP1-FIIND$^{UPA}$ self-oligomerization. In total, nine mutations aimed at disrupting bulky sidechain residues were introduced into a SUMO-NLRP1-FIIND$^{UPA}$ construct based on the homologous UPA structure (PDB 3G5B)[30]. The P1278A P1279G (PP1278AG, colored in red) mutant had the strongest effect at disrupting the oligomerization of SUMO-NLRP1-FIIND$^{UPA}$ on native gel (Fig. 3a), whereas four other mutants (colored in orange), R1240D, RK1260DD, R12785D, and F1320A more subtly reduced NLRP1-FIIND$^{UPA}$ oligomerization. In agreement with the necessity for the UPA domain to facilitate NLRP1 activation, all oligomerization mutants abrogated the ability for NLRP1-FIIND$^{UPA}$-CARD to induce ASC-GFP specks, with PP1278AG having the strongest effect (Fig, 3b, c). Taken together, these findings suggest that NLRP1-FIIND$^{UPA}$ employs one large hydrophobic surface patch centered around a.a. 1278-1279 (red) to self-oligomerize in a "side-by-side" arrangement (Fig. 3d), giving rise to the outer ring (Fig. 3e) that wraps around the CARD core in the FIIND$^{UPA}$-CARD filaments (Fig. 1j). This organization closely resembles the FIINDUPA-dimer structures seen in two recently reported NLRP1-DPP9 complexes[19,21]. A similar architecture is likely true for CARD8 (Fig. 3g). This model also explains the necessity for the interdomain linker between UPA and CARD for optimal NLRP1 and CARD8 filament formation (Supplementary Fig. 3c). A zoomed-in view reveals that the optimal linker between UPA and CARD is ~25~40 Å (Fig. 3f). Shorter linkers would incur spatial constraint and prevent filament extension, whereas longer linkers would produce an outer layer that is too far away from the inner core, potentially destabilizing the FIIND$^{UPA}$-CARD complex. Multiple sequence alignment confirmed that 9–20 a.a. linker length is true for both human and rodent NLRP1 homologs (Supplementary Fig. 2d), suggesting that arrangement of the FIIND and CARD domains are conserved.

**NLRP1 and CARD8-CARD oligomers have distinct surface charge distributions despite similar helical symmetry.** Despite near-identical helical parameters, NLRP1-CARD and CARD8-CARD filaments have remarkably different surface charge distributions (Fig. 4a–f), that are not suggested by their monomer structures alone[31,32]. NLRP1-CARD filaments demonstrate a balanced charge distribution on its axial surface, similar to previously published DD complexes (Fig. 4a)[2,4,33]. By contrast,

in CARD8-CARD filaments, the positively charged patches on the monomer are mostly buried inside the filamentous oligomer, exposing a predominantly negative surface (Fig. 4b). The density from the Type I interface, which has a prominent role in supporting the oligomerization of other CARD helical complexes, does not seem to play a major role in the observed surface density in either complex. Instead, the surface charge distribution is largely due to unusually strong Type II interface densities in both filaments. In the NLRP1-CARD complex, two positively charged histidine residues (His1454 and His1452) extend from the Type IIb surface and interact with negatively charged residues, Asp1437 and Ser1435 on the Type IIa surface (Fig. 4c). In contrast, the CARD8 Type IIb surface is partially negatively charged (Tyr531, Tyr527, and Asp525), whereas the Type IIa surface is positive (Lys508, Lys509, and Leu512) (Fig. 4d). These distinct arrangements of the Type II interfaces are a direct result of the subtle differences (2 degree in helical rotation and ~15 degree of monomer rotational position) in their helical symmetries, and therefore not easily predicted from the amino-acid sequence differences between NLRP1 and CARD8. Interestingly, the top surface of NLRP1-CARD closely resembles the charge distributions of ASC-CARD oligomer top surface (ASC-CARD filament solved in-house, PDB-6K99, EMD-9947, Fig. 4e, g, left), and CARD8 top surface is more similar to CASP1-CARD top surface (Fig. 4f, h, left). Hence, NLRP1-CARD top surface is predicted to preferentially interact with the bottom surface of ASC-CARD (compare Fig. 4e, g, right), whereas the CARD8-CARD top surface is expected to preferentially interact with the CASP1-CARD bottom surface (compare Fig. 4f, h, right). These structural findings offer a potential explanation for the differing affinity and requirement for ASC between NLRP1 and CARD8, which was recently inferred on the basis of cellular experiments.

**Structural basis for the distinct downstream CARD preference between NLRP1 and CARD8.** To identify the precise structural determinants that are responsible for ASC vs. pro-caspase-1 binding, we focused on the Type II surfaces of the respective CARD oligomers. This surface has a prominent role in the observed differences in surface charge distribution between NLRP1-CARD and CARD8-CARD filaments (Figs. 2j, k, 4a–f). We hypothesized that the Type II junctions are not only important for homotypic assembly between identical CARD subunits, but also heterotypic interactions between "sensor" CARDs (NLRP1 and CARD8) and downstream effector CARDs (ASC and CASP1). To test this, we first identified the Type II interfaces in published ASC-CARD and CASP1-CARD filament structures (Supplementary Fig. 6a–e, Supplement Table 1). In both structures, we noted a similar internal density bridge on Type II interface as what was observed in NLRP1-CARD and CARD8-CARD complexes (compare Fig. 5a, b to Fig. 4c, d),

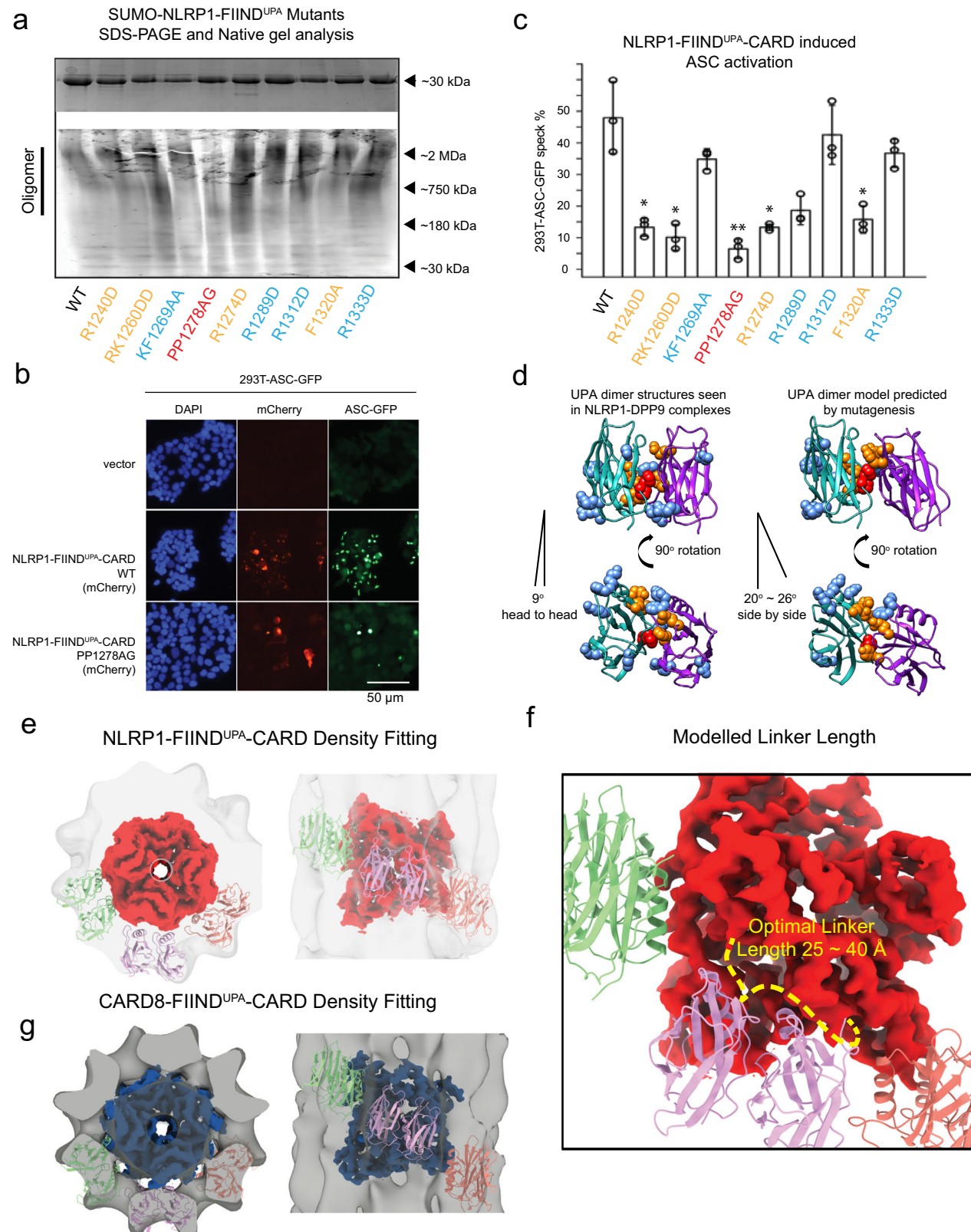

which had not been highlighted by previous analyses[24,29,34–40]. In ASC-CARD filaments, this bridge consists of two residues Gln185 and Tyr187, which extend from the Type IIb surface and interact with Ala168 and Asn170 on the Type IIa surface (Fig. 5a). In CASP1-CARD filaments, the Type II interface bridge employs a distinct IIa surface composed of Lys64 and Gln67, which interacts

with Type IIb residues, Asp80 and Tyr82 (Fig. 5b). This "bridge-like" feature across the Type II interface becomes even more apparent when we refined the existing CASP1-CARD model (Supplementary Fig. 6c). Hence, a strong inter-subunit Type II density bridge seems to be a common feature amongst inflammasome CARD complexes.

**Fig. 3 Model of two-layer organization of NLRP1 and CARD8 FIIND$^{UPA}$-CARD filaments. a** Top panel: SDS-PAGE NLRP1-FIIND$^{UPA}$-CARD mutants. Bottom panel: native gel analysis of SUMO-NLRP1-FIIND$^{UPA}$ constructs, stained by Krypton. **b** ASC-GFP speck formation induced by wild-type NLRP1-FIIND$^{UPA}$-CARD and PP1278AG in 293T-ASC-GFP cells. **c** Percentage of ASC-GFP specks induced by NLRP1-FIIND$^{UPA}$-CARD constructs in 293T-ASC-GFP cells. Cells were fixed 24 hours after transfection for fluorescence microscopy. Data are presented as mean values ± SD. *P* value was calculated with one-way ANOVA, $n = 3$ biological replicates. '*/**/****' indicates *P* value < 0.05, 0.01, and 0.0001, correspondingly. **d** Side view and top view of FIIND$^{UPA}$-dimer structures reported by other groups, and our modeled oligomer interface of NLRP1-FIIND$^{UPA}$. Potential surface residues that may involve homotypic oligomerization were displayed using spheres, PP1278-1279, were highlighted in red color, R1240, RK1260, R12785, and F1320 are colored in orange. **e** The top and side views of NLRP1-CARD filament and six modeled FIIND$^{UPA}$ domains fitted into the FIIND$^{UPA}$-CARD filament density. **f** A close-up view of the potential UPA-CARD linker orientation. A linker with 25~40 Å is likely to support a regular organization of the two-layer structure. **g** The top and side views of CARD8-CARD filament and si modeled FIIND$^{UPA}$ domains fitted into the CARD8-FIIND$^{UPA}$-CARD filament density.

Using the refined assignment of Type II interface residues, we modeled all the pairwise unidirectional CARD-CARD interactions among NLRP1, CARD8, ASC, and CASP1 (Fig. 5c–f), based on our in-house built filament models (PDB-6K7V, 6K9F, 6K99) and newly refined CASP1-CARD oligomer structure (locally refined PDB-5FNA using our CARD8-CARD structure as a reference). A remarkable level of specificity becomes apparent, as certain interactions are favored, whereas others show strong incompatibility. NLRP1-CARD and ASC-CARD are expected to be compatible, as the prominent HPH (a.a. 1452-1454) motif at the NLRP1 Type IIb interface can form ionic interactions with the ASC IIa residues AWN (a.a. 168–170) (Fig. 5c, d, left panel). In contrast, the same NLRP1 HPH motif clashes with the CASP1 Type IIa surface residues, KGAQ (a.a. 64–67) (Fig. 5c, d, right panel, circle). In other words, NLRP1-CARD uses the HPH motif on its Type IIb surface to preferentially engage ASC-CARD and disfavor CASP1-CARD (Fig. 5c, d). On the other hand, CARD8-CARD employs a DPY (a.a. 525–527) motif on its Type IIb surface. This allows CARD8-CARD to interact strongly with the CASP1-CARD Type IIa surface via ionic interactions between Asp 525 (CARD8) and Lys64 (CASP1) (Fig. 5c, f, right panel). In contrast, the same DPY motif lacks effective interaction with the ASC-type IIa surface, as the charged Asp 525 in the CARD8 DPY motif cannot pair with Ala 168 on the ASC-type IIa surface (Fig. 5f, left panel). Hence, CARD8-CARD is structurally compatible with CASP1-CARD but not ASC-CARD. These results demonstrate that NLRP1-CARD and CARD8-CARD have the intrinsic ability to discriminate between ASC and pro-caspase-1 due to the structural features of their Type IIb motifs.

This analysis can be further extended to explain why ASC is uniquely suited to act as an adaptor and amplifier in inflammasome signaling. To do so, ASC-CARD must engage upstream sensors, self-oligomerize, and activate CASP1-CARD simultaneously. This versatility is endowed by its unique Type IIb motif, QSY (a.a. 185–187), which possesses biochemical properties and charges that are intermediate between NLRP1-CARD and CARD8-CARD. The Type IIb QSY motif in ASC-CARD adopts a conformation that is compatible both with its own Type IIa residues, i.e., a.a. AWN (a.a. 168–170), as well as the CASP1 "KGAQ" motif (Fig. 5e). This is achieved with the help of additional sidechain features. For instance, instead of a prominent negative Asp residue in CARD8 Type IIb surface, ASC-CARD uses a less-charged Asn residue to enhance the interaction with Ala168 on its own type IIa surface. In addition, the tyrosine residue (Tyr187) on ASC-CARD Type IIb demonstrate a more inwards orientation as compared to the equivalent tyrosine residue on CARD8 type IIb surface (Tyr527), thus allowing it to effectively engage both its own Type IIa as well as that of CASP1-CARD (Fig. 5e).

To further corroborate that signaling specificity is intrinsic to the CARD domains, we performed an in vitro oligomerization assay using recombinant fluorescently labeled CASP1-CARD and

ASC-CARD. In agreement with the structural predictions, CARD8-CARD, but not NLRP1-CARD, preferentially stimulated CASP1-CARD oligomerization, as demonstrated by the appearance of slower migrating bands on native gels (Supplementary Fig. 6f, left panel, lanes 2 and 3 vs. 4 and 5), whereas the reverse was true for ASC-CARD oligomerization (Supplementary Fig. 6f, right panel, lanes 2 and 3 vs. lanes 4 and 5). Next, we used structured illumination microscopy (3D-SIM) to directly visualize NLRP1-CARD and CARD8-CARD and their interactions with downstream CARDs (ASC-CARD vs. CASP1-CARD) within cultured human cells. Both mCherry-tagged NLRP1-CARD and CARD8-CARD formed filament bundles within the cytosol (Fig. 6a, red), which intertwined with those formed by GFP-tagged ASC-CARD and CASP1-CARD, respectively (Fig. 6a, green). Given the theoretical resolution of ~110 nm in width (*x–y* plane), we postulated that the thinnest filament bundle observed was no more than approximately six times the width of an individual filament with the mCherry tag. In many instances, ASC-CARD or CASP1-CARD filaments could be observed as "emanating: from one side of NLRP1-CARD or CARD8-CARD filament bundles (Fig. 6a). This "filament-seeding" phenomenon was not limited to the isolated CARDs, as Talabostat-activated full-length NLRP1 and CARD8 also formed filaments[9,15,22] and colocalized with ASC-CARD and CASP1-CARD filaments, respectively (Supplementary Fig. 7). Furthermore, Talabostat-activated full-length NLRP1, or its activating fragment NLRP1-FIIND$^{UPA}$-CARD are unable to trigger CASP1-GFP filaments in the absence of ASC, whereas CARD8 (or CARD8-FIIND$^{UPA}$-CARD) was unable to nucleate ASC-GFP specks in 293 T cells (Supplementary Fig. 8a). Taken together, these results provide direct cellular evidence that NLRP1-CARD and CARD8-CARD form filament-like oligomers that differentially engage either ASC or pro-caspase-1, via heterotypic CARD interactions.

We further corroborated that CARD8 is the first human inflammasome sensor that is unable to engage ASC as an amplifier[18]. In immortalized human keratinocytes, CRISPR/Cas9-mediated deletion of endogenous *ASC* (*PYCARD*) abrogated the ability of overexpressed NLRP1-FIIND$^{UPA}$-CARD fragment or NLRP1-CARD to induce endogenous IL-1β secretion (Fig. 6c, red bars, Supplementary Fig. 8c–e) and pyroptotic cell death (Fig. 6d, red bars, Supplementary Fig. 8b, d). In contrast, CARD8-dependent pyroptosis was not abrogated in *ASC* (*PYCARD*) KO keratinocytes (Fig. 6c, d, blue bars, Supplementary Fig. 8c, e). Thus, NLRP1 requires endogenous ASC for inflammasome activation in human cells, whereas CARD8 can directly activate caspase-1 independently of ASC.

Finally, we performed a classical "seeding" experiment where purified untagged NLRP1-CARD or CARD8-CARD (the seed) was incubated with the respective SNAP-tagged partner CARD, i.e., ASC-CARD or CASP1-CARD. In this setup, the downstream ASC-CARD or CASP1-CARD filaments appear thicker due to the SNAP tag, and could therefore easily be distinguished from the upstream filaments composed of

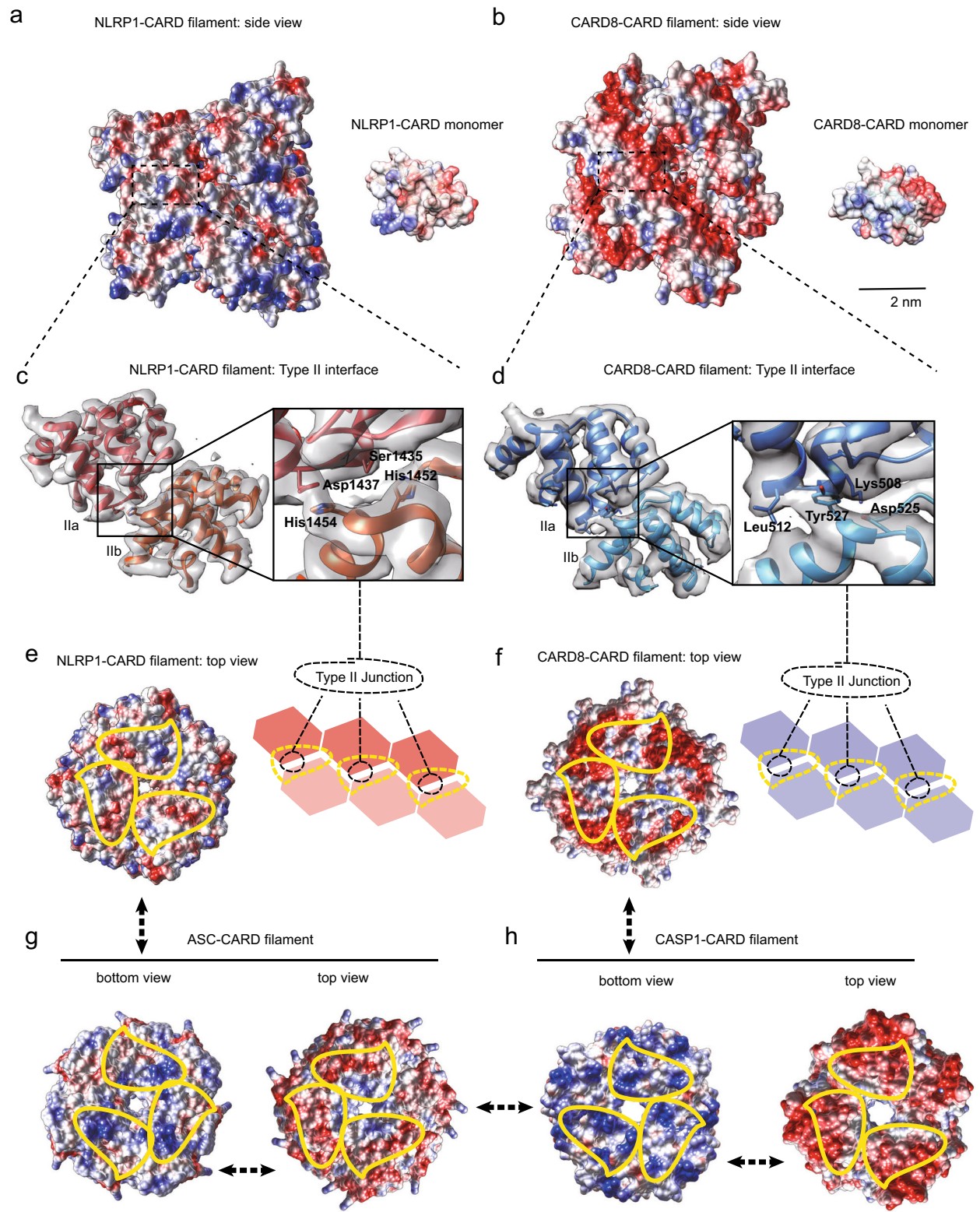

**a** NLRP1-CARD filament: side view  NLRP1-CARD monomer

**b** CARD8-CARD filament: side view  CARD8-CARD monomer

2 nm

**c** NLRP1-CARD filament: Type II interface

IIa
IIb

Ser1435
Asp1437  His1452
His1454

**d** CARD8-CARD filament: Type II interface

IIa
IIb

Lys508
Leu512  Tyr527  Asp525

**e** NLRP1-CARD filament: top view

Type II Junction

**f** CARD8-CARD filament: top view

Type II Junction

**g** ASC-CARD filament

bottom view    top view

**h** CASP1-CARD filament

bottom view    top view

NLRP1- or CARD8-CARD. In several independent experiments, the thick downstream CARD (ASC-CARD and CASP1-CARD) filaments could be seen to extend from one end of the thin NLRP1 and CARD8 upstream seeding filaments (Supplementary Fig. 9a, b). This is consistent with a structural model where oligomerized NLRP1-CARD and CARD8-CARD use a single type II interface to seed ASC-CARD or CASP1-CARD in a unidirectional manner.

**NLRC4-CARD can engage either ASC-CARD or CASP1-CARD**. Our results thus far demonstrate that subtle differences in the inflammasome CARD domain structures, particularly at the type II junctions, can directly dictate the signaling specificity of the NLR sensor proteins. To extend these results beyond NLRP1 and CARD8 and test the predictive power of our model, we examined a third CARD-containing NLR sensor, NLRC4. To unambiguously assign the Type II interface residues of human

**Fig. 4 Oligomeric interfaces on inflammasome CARD domains. a** Left panel: side view of electrostatic potential distribution of a segment of NLRP1-CARD filament (red, −10 C/m²; blue, 10 C/m²; generated in Chimera). Right panel: electrostatic potential map of a NLRP1-CARD monomer in the same orientation as the filamentous oligomer. **b** Left panel: side view of electrostatic potential distribution of a segment of CARD8-CARD filament (red, −10 C/m²; blue, 10 C/m²). Right panel: electrostatic potential map of a CARD8-CARD monomer in the same orientation. **c** Atomic model cartoon and the directly observed density map of a NLRP1-CARD dimer within the filamentous complex, centered at the Type II junction interface. The four residues, His1452-His1454 on the Type IIb interface and the Ser1435-Asp1437 on the Type IIa interface were highlighted and shown in the "stick" mode. **d** Atomic model cartoon and the directly observed density map of CARD8-CARD dimer in the filament, centered at the Type II junction interface. The four residues, Asp525–Tyr527 on the Type IIb interface and the Lys508–Leu512 on the Type IIa interface were highlighted and shown in the "stick" mode. **e** Electrostatic potential map of the top view (Type IIb interface, highlighted in yellow dashed area) of the NLRP1-CARD filament. **f** Electrostatic potential map of the top view (type IIb interface, highlighted in yellow dashed area) of CARD8-CARD filament. **g** Electrostatic potential map of the bottom (type IIa highlighted in yellow dashed area) and top (type IIb highlighted in yellow dashed area) views of the ASC-CARD filament. **h** Electrostatic potential map of the bottom (type IIa highlighted in yellow dashed area) and top (type IIb highlighted in yellow dashed area) views of the CASP1-CARD filament.

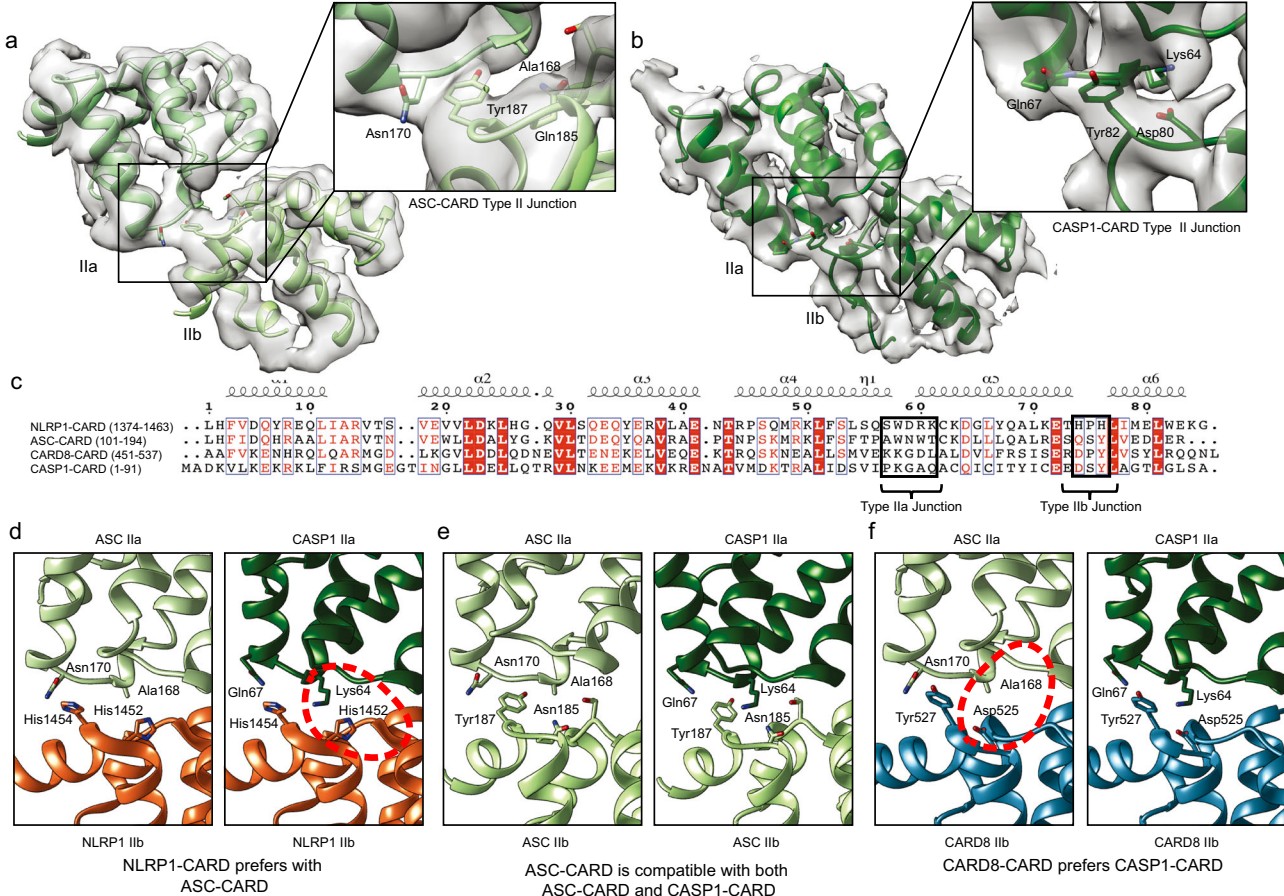

**Fig. 5 Type IIa–b interface influences human inflammasome complexes. a** Type II interface in ASC-CARD filaments (PDB-6K99, EMD-9947). **b** Type II interface in CASP1-CARD filaments after refitting. See Fig. S5c for a comparison between the refitted model and the original model, PDB-5FNA, and EMD-3241). **c** Sequence alignment of human NLRP1-, ASC-, CARD8-, and CASP1-CARD. Important residues within the Type II interfaces are highlighted in black box. **d** Structural compatibility of human NLRP1-CARD with ASC-CARD or CASP1-CARD. Potential clashes are highlighted in red. **e** Structural compatibility of human ASC-CARD with itself or CASP1-CARD. **f** Structural compatibility of human CARD8-CARD with ASC-CARD or CASP1-CARD. Potential clashes are highlighted in red.

NLRC4-CARD, we determined its filament structure at 3.3 Å using cryo-EM (Supplementary Fig. 6d, e). The final polished structure closely agrees with previously published findings[24,39]. Interestingly, unlike NLRP1-CARD and CARD8-CARD, the strong density bridge in the Type II interface is absent within NLRC4-CARD filaments (Supplementary Fig. 10a). The Type IIb tyrosine residue (Tyr78) is folded backward toward the hydrophobic core and does not participate in intermolecular Type II interaction. Instead, two nearby small residues, Asn77 and Gln82, come forward to form a class of Type II junction (Supplementary Fig. 10a, b) that is more flexible than that of NLRP1 and CARD8.

Molecular docking predicted that NLRC4 Type IIb surface would be structurally compatible with both ASC-CARD and CASP1-CARD. This was validated using the in vitro EMSA assay (Supplementary Fig. 6f, lanes 6–7 in both panels) and in 293 T cells. Overexpressed NLRC4-CARD nucleated the formation of both ASC-GFP and CASP1 C285G-GFP specks, as well as ASC-CARD-GFP and CASP1-CARD-GFP filaments at similar levels (Supplementary Fig. 10c, d). The versatility of human NLRC4-CARD is in agreement with previous studies on murine Nlrc4. In mice, *Asc/Pycard* knockout attenuates, but does not abolish caspase-1-dependent pyroptotic cell death[41], suggesting that

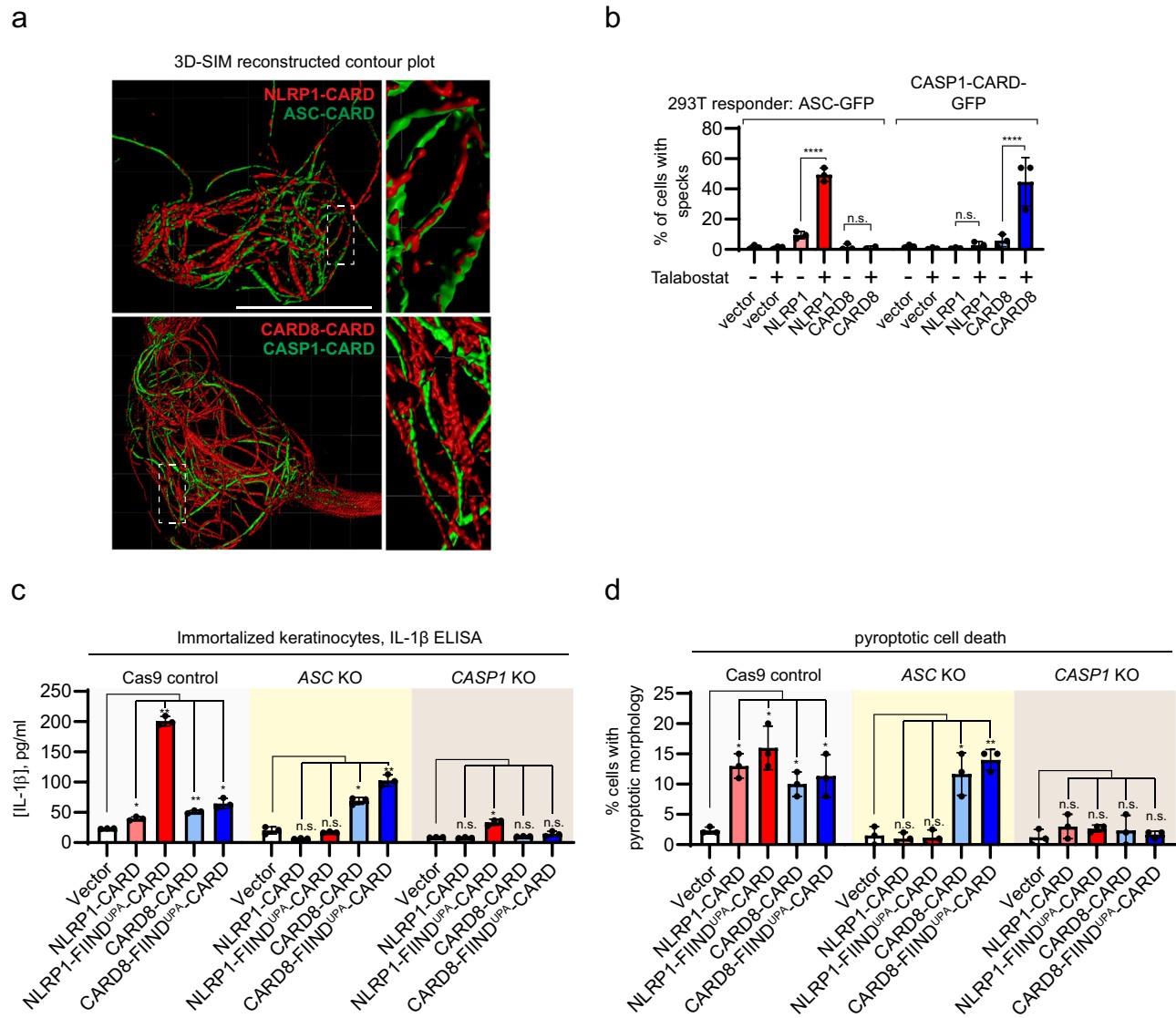

**Fig. 6 NLRP1 and CARD8 inflammasome assembly in mammalian cells. a** Structured illumination microscopy images of NLRP1-CARD-mCherry and CARD8-CARD-mCherry filaments (red) with ASC-CARD-GFP and CASP1-CARD-GFP filaments (green). White boxes indicate regions magnified in right panels. **b** Percentage of ASC-GFP specks or CASP1-CARD-GFP filaments induced by full-length NLRP1 or CARD8 upon Talabostat treatment (3 μM). Data are presented as mean values ± SD. P value was calculated with one-way ANOVA, n = 3 treatments. '*/**/****' indicates P value <0.05, 0.01, and 0.0001, correspondingly. **c** IL-1β secretion in control, *ASC* KO, and *CASP1* KO keratinocytes transfected with the indicated NLRP1 and CARD8 constructs. Data are presented as mean values ± SD. P value was calculated with one-way ANOVA, n = 3 transfections. '*/**/****' indicates P value <0.05, 0.01, and 0.0001, correspondingly. **d** The percentage of cells with pyroptotic morphology in control, *ASC* KO, and *CASP1* KO keratinocytes transfected with the indicated NLRP1 and CARD8 constructs. Data are presented as mean values ± SD. P value was calculated with one-way ANOVA, n = 3 transfections (same as 5c). '*/**/****' indicates P value < 0.05, 0.01, and 0.0001, correspondingly.

murine NLRC4, similar to its human homolog can engage both ASC and pro-caspase-1 (Supplementary Fig. 10e).

## Discussion

Taken together, our cellular, biochemical, and structural findings suggest a model for NLRP1- and CARD8 inflammasome activation (Fig. 7). We propose that FIIND auto-cleavage, which has been shown previously to be required for NLRP1 activation, serves as a necessary "licensing" step to liberate the FIIND$^{UPA}$ domain. Upon ligand-induced or 3C protease-mediated[42] activation and relief of auto-inhibition, FIIND$^{UPA}$ self-assembles into ring-like oligomers in a "head-to-tail" fashion (Fig. 3), bringing adjacent CARD monomers into close proximity. This lowers the threshold CARD oligomerization and allows for further

"prionoid"-like growth of FIIND$^{UPA}$-CARD filaments (Fig. 7a). Our model of CARD8-FIIND$^{UPA}$-CARD suggests that this model likely also holds true for the CARD8 inflammasome (Fig. 7b).

We next solved the cryo-EM structures of NLRP1-CARD and CARD8-CARD filaments. Despite very similar helical parameters, the subtle differences in rotational angles of individual NLRP1-CARD and CARD8-CARD monomers and their type II junctions give rise to distinct surface charge distributions on the filament scale. Extending this analysis, we also found that specific type II interface motifs directly dictate the compatibility of heterotypic pairwise CARD-CARD interactions among NLRP1, CARD8, NLRC4, ASC, and CASP1. In particular, these results explain why NLRP1, but not CARD8, requires ASC for inflammasome activation. Thus, the CARD domains are not mere interchangeable "lego building blocks". Instead, each possesses intrinsic abilities to

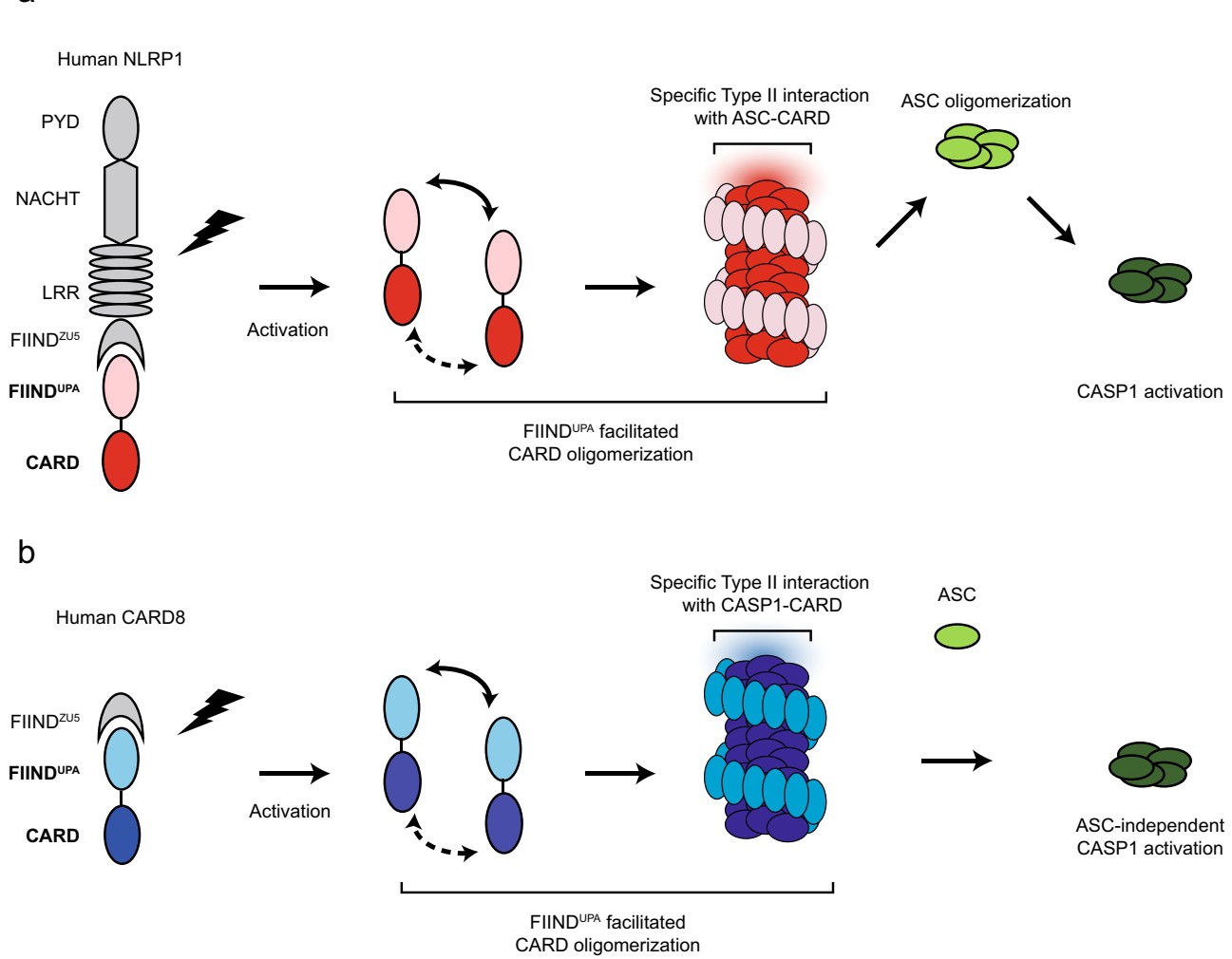

**Fig. 7 A two-step model for human NLRP1 and CARD8 inflammasome assembly.** From left to right, NLRP1 (**a**) and CARD8 (**b**) are "licensed" by auto-proteolysis within the FIIND. Upon activation by PAMP or DAMP triggers, the self-inhibition exerted by the N-terminal proteolytic fragment is relieved. FIIND^UPA facilitates the oligomerization of CARD, leading to the formation of a two-layered filament complex. Oligomerized NLRP1-CARD and CARD8-CARD differentially engage ASC or CASP1, leading to the assembly of distinct inflammasome complexes.

discriminate amongst potential downstream binding partners in order to specify the composition of different inflammasome complexes. It is conceivable that these distinct inflammasome complexes can lead to distinct biological outcomes, in agreement with previous findings demonstrating that different strengths of the pyroptotic signaling are required for cell death and IL-1 cytokine secretion[41,43]. Interestingly, the prominent role in type II interface-dependent assembly is apparently specific to the inflammasome complexes, as other CARD domain innate immune complexes, such as MAVS (EMD-5925)[33], BCL10 (EMD-7314)[44], RIPK2 (EMD-4399, 6842)[35,45] rely more heavily on Type I interface junctions (e.g., MAVS Arg43-Typ56 density bridge). This underscores the importance of studying molecular structures of large complexes at atomic resolutions. As supra-molecular assemblies are highly prevalent in innate immune and cell death signaling[46], our findings should inform future work to decipher the general assembly principles of other molecular complexes involved in these vital processes.

## Methods

**Cell culture**. HEK293Ts (ATCC #CRL-3216) were obtained from commercial sources and cultured according to the suppliers' protocols. Immortalized human keratinocytes (N/TERT-1) were a kind gift from H. Reinwald (MTA)[47]. All cell lines underwent routine Mycoplasma testing with Lonza MycoAlert (Lonza #LT07-118).

**Transient transfection and stable cell line generation using lentiviruses**. 293T-ASC-GFP cells were described previously[8]. All other 293T-GFP responder cell lines were constructed by lentiviral transduction (pCDH vector backbone, System Biosciences). All transient expression plasmids were cloned into the pCS2+ vector using standard restriction cloning. Polyclonal Cas9/CRISPR knockout cell lines were generated with lentiCRISPR-v2 (Addgene #52961) and selected with puromycin. Knockout efficiency was tested with western blot 7–10 days after puromycin selection. Site-directed mutagenesis was carried out with QuickChangeXL II (Agilent #200522).

**3D-SIM and widefield-deconvolution**. A DeltaVision OMX v4 Blaze microscope (GE Healthcare) equipped with 488 nm and 568 nm lasers for excitation (3D-SIM) and solid-state illuminator (widefield) the BGR-FR filter drawer (emission wavelengths 436/31 for DAPI) was used for acquisition of 3D-SIM and widefield-deconvolution images. An Olympus Plan Apochromat ×100/1.4 point spread function oil immersion objective lens was used with liquid-cooled Photometrics Evolve EM-CCD cameras for each channel. For 3D-SIM, 15 images per section per channel were acquired (made up of three rotations and five-phase movements of the diffraction grating) at a z-spacing of 0.125 μm. The same z-spacing was used for widefield acquisition. Structured illumination reconstruction or deconvolution and alignment was completed using the SoftWorX (GE Healthcare) program, 3D image rendering and analysis in Imaris version 9.3.0 (Bitplane an Oxford Instruments

Company), 2D image analysis in Fiji (Schindelin et al., 2012) and figure preparation in Illustrator.

**Antibodies and cytokine analysis**. The following antibodies were used in this study: HA tag (Santa Cruz Biotechnology, #sc-805, 1:1000 for western blot), GAPDH (Santa Cruz Biotechnology, #sc-47724, 1:1000 for western blot), ASC (Adipogen, #AL-177, 1:1,000 for western blot), CASP1 (Santa Cruz Biotechnology, #sc-622, 1:1000 for western blot), IL-1B (R&D systems, #AF-201, 0.1 μg/mL for western blot, 2 μg/mL for immunochemistry), FLAG (SigmaAldrich, #F3165, 1:2000 for western blot), GFP (Abcam, #ab290, 1:2,000 for western blot). IL-1B measurements were carried out with human IL-1B ELISA kit (BD, #557953, follow supplier instructions).

**Plasmid construction for recombinant protein expression**. For recombinant CARD proteins, NLRP1 constructs (985-1473, 1213–1473, 1374-1466, 1374–1473) with or without C-terminal SNAP tag (NEB), CARD8-CARD (451–537) with or without C-terminal SNAP tag, ASC-CARD (1–102) with N-terminal SNAP tag, CASP1-CARD (1–92) with or without C-terminal SNAP tag, NLRC4-CARD (1–102) with or without C-terminal SNAP tag. were cloned into pET47 vector in between XmaI and XhoI restriction sites. A 3 C protease cleavage site was inserted in between the CARD domain and the SNAP tags in case SNAP needs to be removed for biochemical analysis. Site-directed mutagenesis was performed using the KAPA Hifi PCR kits (KAPA Biosystems).

**Protein expression and purification**. For the expression of the His-tagged recombinant proteins, the constructed plasmids were transformed into BL21(DE3) (NEB) and grown at 37 °C for 16–18 h. Then the starting culture was transferred to a larger volume of lysogeny broth, which was grown until an OD600 of ~0.6-0.8, then the temperature was lowered to 16 °C, and the cells were induced with 0.5 mM IPTG and grown overnight. Bacteria were harvested at $4600 \times g$ (Avanti JXN series, Beckman Coulter) for 10 min and then resuspended with lysis buffer (20 mM Tris-HCl, pH 8.0, 300 mM NaCl, 20 mM imidazole, and 10% glycerol). Cells were then lysed by Emulsiflex-C3 and centrifuged at a speed of $30,000 \times g$ (Avanti JXN series, Beckman Coulter) for 20 min at 4 °C to separate supernatant and pellet. The supernatant was passed through a pre-equilibrated column containing Ni-NTA agarose beads (ThermoFisher Scientific) by gravity. The column was then washed with lysis buffer containing 20 mM imidazole to remove nonspecific binding proteins. His-tagged proteins were eluted with elution buffer that contained 300 mM imidazole. To further purify our recombinant proteins, size-exclusion chromatography (Superdex 200 increase, GE Healthcare) was performed in buffer A (20 mM Tris-HCl, pH 8.0, 150 mM NaCl, and 1 mM DTT). At this step, the purified proteins were the oligomeric pre-activated seed fraction. For monomeric CARD-SNAP or SNAP-CARD domain proteins, it was prepared following the protocol as described here. Concentrated and purified soluble filament seed at around 10 mg/ml was buffer exchanged into 10 ml of 6 M guanidinium, and gradually dialyzed against 3, 2, 1.5, 1, 0.8, and 0.6 M guanidinium containing buffer A in the cold room, and eventually in buffer A. In the absence of stimuli, refolded SNAP-ASC-CARD protein remained monomeric for 24 h, and CASP1-CARD-SNAP remained monomeric or lower oligomeric for about 12 h. Recombinant SNAP-NLRP1-FIIND$^{UPA}$-CARD and SNAP-CARD8-FIIND$^{UPA}$-CARD were expressed in similar ways as the CARD domain constructs. Notably, most of the protein exist in cellular pellets after the Emulsiflex step, thus, direct pellet wash with lysis buffer with an additional 1% Triton-X100, followed by 6 M Guanidinium is necessary to obtain solution proteins. Chemical refolding following the same dialysis protocol, followed by 3 °C cleavage of the SNAP tag, was used to obtain homogenous oligomeric population of FIIND$^{UPA}$-CARD complexes at various concentrations.

**In vitro oligomerization assay**. SNAP-ASC-CARD and CASP1-CARD-SNAP monomers that were labeled with Alexa Fluor 647 (New England Biolabs) were co-incubated with CARD domain only filament seeds (CARD8-CARD without SNAP, NLRP1-CARD without SNAP, NLRC4-CARD without SNAP) separately. One microliter of 100 μM BG-Alexa dye stocks (dissolved in water containing 5% dimethyl sulfoxide) were added into every 100 μl of pre-labeling protein stock at a concentration of 20 μM. Fluorescently labeled monomeric SNAP-ASC-CARD (10 μM) and CASP1-CARD-SNAP (10 μM) was mixed with 20 or 5 μM seed oligomer for 15 min at room temperature before analysis by Bis-Tris native PAGE gel (Life) or 15% sodium dodecyl sulfate polyacrylamide gel electrophoresis gel. The fluorescently labeled gel was run at a voltage of 200 V for 60 min. Once the EMSA was completed, they were visualized by Typhoon FLA7000 (GE Healthcare) with 100-μm pixel size at channel 647 using PTM 600. To form ASC-CARD and NLRC4-CARD long filament (200–500 nm), ~20 μM monomeric SNAP-tagged protein was mixed with unlabeled, wild-type SNAP-tagged ASC-CARD and NLRC4-CARD seeds at a molar ratio of 20:1. Later on, in-house-prepared MBP-3C-protease was added to the fully extended filament with a final concentration of 0.5 μM. The digestion was performed at 4 °C for 8 h, before further size-exclusion purification to separate the core filament, the cleaved-off SNAP tags, and proteases. Final extended ASC-CARD and NLRC4-CARD thin filament were then purified using size-exclusion column Superose 6 (GE Healthcare), before structural analysis. To form long NLRP1-CARD and

CARD8-CARD filaments, N-terminal His-tagged CARD domain alone constructs were expressed and purified according to the protocol mentioned above, then concentrated to ~400 μM, homogeneous long filaments formed spontaneously, without the requirement of seeding.

**Negative stain EM**. Samples were applied onto the grids and stained following a Harvard medical school Liao lab protocol (https://liao.hms.harvard.edu/node/32, originally designed by Tom Walz). Protein samples were diluted to 0.01–0.05 mg/ml for best sample density on the grids, immediately before grid preparation to prevent concentration-dependent dissociation. Images were taken using a T12 120 keV TEM microscope, at either 30,000× or 48,000× magnification, and calibrated against known objects of MAVS-CARD filament.

NLRP1-FIIND$^{UPA}$-CARD complex density model was built based on 58 negative stained EM images with homogeneous background, total 818 segments from a single 15 nm thick 2D class. Combining more segments yielded very noisy maps. Several preliminary maps ~29 Å were obtained after refinement, when different helical parameters were adopted, but all maps have the similar two-layer features. In addition, such strong inner layer density and spotted outer layer density are always seen in all solutions. The illustrated 3D map was obtained when choosing helical parameters similar to NLRC4 density EMDB-2901 and analysis of the NLRP1-FIIND$^{UPA}$-CARD 2D class images. We were cautious about the accuracy of this particular map, thus simply using it as a guide to piece together an overall architecture of NLRP1 inflammasome.

**Cryo-EM and helical reconstruction**. For cryo-EM grid preparation, Quantifoil R1.2/1.3 holey grid (Quantifoil) was glow-discharged for 60 s and plunge-frozen using a Vitrobot (FEI) to be treated with 5-μl sample at a concentration of ~2 mg/ml. Final high-resolution images were collected at Titan Krios, 300 kV, using Gatan K2 camera (Gatan), super-resolution mode for data collection, and binned 2× when doing motion correction. The sub-frame time is 250 ms and total exposure time is 10 s, this will give 40 frames per stack. The pixel size on the final image is 1.10 Å. The dose rate is 4 e/pixel/s.

For cryo-EM data, super-resolution image stacks were "gain-normalized" and binned by 2× to a pixel size of 1.10 Å prior to drift and local movement correction using MotionCorr2. Date processing software Relion 3.0 was used for CTF determination, particle picking, 2D classification, 3D classification, and refinement procedures. Totally, 2187 (NLRP1), 1424 (CARD8), 3183 (ASC), and 2109 (NLRC4) micrographs were collected and analyzed. We manually picked out ~10,000 to 20,000 filaments for each sample and windowed out segments of 240 × 240 pixels, yielding 422,388 (NLRP1), 1,201,108 (CARD8), 877,406 (ASC-CARD), and 382,715 (NLRC4) particles for 2D classification. After several iterations, bad particles and less desirable classes were removed and we used 100,240 (NLRP1), 301,848 (CARD8), 131,260 (ASC-CARD) and 120,602 (NLRC4) particles for 3D classification. Ab initio low-resolution helical structure was generated using a Gaussian cylinder as an initial model. 3D classification and auto-refinement of 53,241 (NLRP1), 98,280 (CARD8), 59,566 (ASC-CARD), and 120,602 (NLRC4) particles were done in RELION 3.0 and CisTEM. All 3D refinements were carried out in RELION. After refinement was converged, a mask of the central 30% segment was calculated and applied to the final data set and the corrected Fourier shell correlation (FSC) was calculated to estimate the resolution ~3.7 Å (NLRP1), 3.7 Å (CARD8), 4.1 Å (ASC-CARD), 3.3 Å (NLRC4) by using FSC = 0.143 criterion. In the final model, parameters of the RIP2 filament were calculated to yield 5.36 Å (NLRP1), 5.409 Å (CARD8), 5.27 Å (ASC-CARD) and 5.17 Å (NLRC4) of the axial rise per asymmetric unit and an azimuthal rotation per subunit of −100,821° (NLRP1), −99,16° (CARD8), −100,614° (ASC-CARD), −100,55° (NLRC4). Helical reconstruction parameters were determined using a real-space exhaustive parameter search method. About 1,600 combinations of helical parameters were applied to each data set and the output density was independently scored based on their biochemical properties. The simulated power spectrum of assigned helical parameters were also compared with the actual 2D images to confirm the accuracy of helical parameter search (Supplementary Fig. 11).

**Model building**. Unpolished maps were used for model building.

Step 1 was the monomer docking. The CARD monomer structures (PDB-4IFP for NLRP1-CARD, PDB-4KIM for CARD8-CARD, PDB-6N1H for ASC-CARD, and PDB-6N1I for NLRC4-CARD) were used as starting points at this stage. In the cases of NLRP1 and CARD8, the residue numbers were renumbered and the terminal residues were deleted to match the length of the density. The individual helix was manually shifted (rigid body movement) according to the locations of three large aromatic side chains. The density map at this resolution enabled us to unambiguously locate these residues. This step was done in Coot manually.

In Step 2, a tetramer containing the Type I, II, and III interfaces (a centrally located monomer and three other monomers forming Type I, II, and III interfaces with the central one) was built. After building four monomers in this way, we obtained a tetramer that could be used to refine the surface interactions. Phenix real-space refinement was done without rigid body refinement. Different parameters were tested, a torsion-rotational angle NCS constraints (Phenix default auto setting) was used at this step. In this way, all surfaces of the central monomer could be refined.

During Step 3, a tetramer model was built that is in agreement with the data from the cellular activity assays, given particular emphasis in charge-reversal experiments.

In Step 4, a 12mer model was built. The central monomer from Step 3 was copy-and-pasted to occupy the entire 60 Å segment density. In the end, 12 identical monomers were added to the density. Phenix real-space refinement with default NCS torsion-rotational constraint was performed. After several rounds of rigid body refinement, a model with reasonable statistics was obtained.

Step 5: Final refinement. Specific issues, like high clash score and Ramachandran plot outliers after Step 4 were manually identified and rebuilt and then Step 4 was repeated until satisfactory statistics and density interpretation were achieved.

ChimeraX was used to produce illustrations of atomic densities and molecular structure models.

**Statistics and reproducibility**. All negative stain EM experiments were reproduced at least two times using different batches of biological material. In particular, for material/constructs that were eventually used for structural analysis and determination (including NLRP1-FIIND$^{UPA}$-CARD, CARD8-FIIND$^{UPA}$-CARD, NLRP1-CARD, CARD8-CARD, ASC-CARD, NLRC4-CARD), at least four different batches of biological samples (independently expressed, purified, and seeded) were analyzed. All presented images were representative observations from these repeated experiments. Biochemical and cellular experiments were reproduced using at least three independent biological samples.

**Reporting summary**. Further information on research design is available in the Nature Research Reporting Summary linked to this article.

## Data availability

The densities and structure models discussed in this study were validated and deposited in public databases. The accession codes and links to these files are: NLRC4-CARD filaments (PDB-6K8J, EMD-9946); ASC-CARD filaments (PDB-6K99, EMD-9947); NLRP1-CARD (PDB-6K7V, EMD-9943); CARD8-CARD (PDB-6K9F, EMD-9948). Other data are available from the corresponding authors upon reasonable request. Source data are provided with this paper.

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

## Acknowledgements

We thank all members of the Zhong laboratory, Reversade laboratory, and Wu laboratory for their support and help. We acknowledge useful discussions with Dr. Luo Dahai (NTU, Singapore) and Dr. Seth Masters (WEHI, Australia). This work is supported by the Agency for Science, Technology and Research (GODAFIT Strategic Positioning Fund, Bruno Reversade), National Medical Research Council, Singapore (NMRC/OFYIRG/0046/2017, Franklin Zhong), Concern Foundation's Conquer Cancer Now Award (Franklin Zhong), National Research Foundation, Singapore (Franklin Zhong), Nanyang Assistant Professorship (Wu Bin, Franklin Zhong), MOH-NMRC-OFIRG grant (MOH-000382-00) and Tier II Ministry of Education Research Grant (MOE2016-T2-1-010-Wu Bin), as well as Tier I grants (2018-T1-002-010). Bruno Reversade is a fellow of the Branco Weiss Foundation and a recipient of the A*STAR Investigatorship, Senior NRF investigatorship and EMBO Young Investigatorship. Franklin Zhong and Wu Bin are Nanyang Assistant Professors. Franklin Zhong is a recipient of the National Research Foundation Fellowship.

## Author contributions

Q.G. prepared NLRP1 protein samples, conducted all in vitro biochemical experiments, prepared EM samples, collected EM data, and analyzed the results. K.R. and D.T. carried out all cellular experiments and microscopy. C.X. prepared NLRC4 protein samples, conducted imaging processing and computational work. J.Z. prepared a CARD8 sample and conducted computational work. Z.Z.B. prepared ASC proteins conducted biochemical assays related to ASC. P.T.H., H.C.C., and Y.J.T. participated in determining CARD8-FIIND$^{UPA}$-CARD filament architecture. Y.Z. facilitated all protein expression and participated in cloning work. K.R., J.L., W.I.G., and G.W. jointly conducted 3D-SIM experiments and data analysis. K.L. participated in cellular experiments. B.W. and C.X. performed cryo-EM data processing. F.Z., B.W., and B.R. jointly conceived of the project, designed and coordinated the experiments, analyzed the results and wrote the manuscript.

## Competing interests

The authors declare no competing interests.
