## [Peer Review File · Nature Communications]

Reviewers' Comments:

Reviewer #1:

Remarks to the Author:

In this work the authors investigate the structural basis of NLRP1- and CARD8-mediated inflammasome signalling using a combination of structural and cell-based approaches. They solved the filamentous structures of NLRP1-CARD and CARD8-CARD, and found that the FIINDUPA preceding the CARD is prone to oligomerization and triggers the functional FIINDUPA-CARD fragment of NLRP1 and CARD8 forms an ordered two-layer filament. They also investigated the potential structural mechanism by which NLRP1 and CARD8 could discriminate between ASC and pro-caspase-1, resulting distinct downstream signalings. This work has potential to be of broad interest in the innate immune field. However, there are still several points need to be addressed before the further consideration of their manuscript.

Major:

1. FIINDUPA-CARD of NLRP1 and CARD8, upon unleashed in response to activating signals, is actually the functional fragment which assembles into oligomeric inflammasome. As the authors showed, FIINDUPA is prone to self-oligomerize and likely bring the following CARD into proximity to form filamentous structure. This results in a regular two-layer filamentous assembly of FIINDUPA-CARD. The spiral assembly of FIINDUPA around the central CARD filament is supposed to be built on considerable homotypic interactions between neighboring FIINDUPA. The authors are encouraged to dissect the FIINDUPA interactions which are of great importance to uncover the precise molecular mechanism that initiates the inflammasome assembly of NLRP1 and CARD8.
2. There is no evidence to show that the ring-like oligomers formed by NLRP1 FIINDUPA alone recapitulate the organization and detailed interactions of FIINDUPA as in the spiral assembly of NLRP1 FIINDUPA-CARD filament. The heterogeneity of FIINDUPA oligomers only supports the intrinsic oligomerization property of FIINDUPA and cannot be simply applied to explain the spiral assembly of FIINDUPA in FIINDUPA-CARD filament. Thus, the observation that the oligomers formed by FIINDUPA alone and the outer layer of FIINDUPA-CARD filaments share a roughly similar size should not be conclusive.
3. The authors should examine the topology of recombinant CARD8 FIINDUPA-CARD filaments at 3D-reconstruction level to demonstrate the similar two-layer filamentous assembly before they conclude a common model for both NLRP1 and CARD8 assembly.
4. Experiment evidences are required to show that NLRP1-CARD and CARD8-CARD nucleate respectively ASC-CARD and CASP1-CARD filament assembly unidirectionally. This is the prerequisite that NLRP1-CARD and CARD8-CARD filaments utilize their top surfaces to prime the filamentous assembly of ASC-CARD and CASP1-CARD, respectively.
5. The complementarity of surface charge distribution is lack of precision to well explain why NLRP1-CARD and CARD8-CARD filaments nucleate different downstream CARD. Based on the surface charge distribution shown by the authors, ASC-CARD filament would not prefer to engage CASP1-CARD, which is evidently not the fact. Considering that all the structurally characterized CARD filaments share highly similar helical symmetry parameters, it is likely that the priming CARD filament would elongate by incorporating downstream CARD via the same organization mode. Thus, inspecting the compatibility of all the three types of interface between the primer and downstream CARD is necessary.
6. The highlighted dominant role of type II interface in discriminating downstream CARD by NLRP1-CARD and CARD8-CARD should be proved by mutagenesis analyses, including loss of preformed intermolecular interactions such as NLRP1-ASC or CARD8-CASP1, and gain of nonexistent intermolecular interactions such as NLRP1-CASP1 or CARD8-ASC.

Minor:

- 1) Introduction, line 65, CARD8 is conceptually not a NLR protein, but a FIIND domain-containing protein closely related to NLRP1 which can activate ASC-independent inflammasome.
- 2) Considering that the filamentous structure of PYRIN or CARD domain was well studied before and likely a universal mechanism for inflammasome assembly, this documented background should be referred to in the introduction part.

- 3) Figure 1e, the images of negative-stain EM are of less quality than required. In the field shown, 100 μ M of NLRP1 FIINDUPA-CARD does not mainly form long filament by self-assembly; also the low-order oligomerization of FIINDUPA-CARD at 10 μ M could not be distinguished from the supposed monomeric CARD at 100 μ M. The author should explain how they estimate the size of the oligomers.
- 4) Line 131-132, comparison of the oligomers formed by diverse NLRP1 fragments would be better demonstrated by back-to-back negative-stain images for all the fragments (with the same scale bar). In the size-exclusion profiles such as figure S1c and S1d, the molecular weight standard should be indicated.
- 5) The authors should clarify whether figure 1g is the representative 2D classification for cryo-EM or negative-stain EM raw images. The description in the text (line 137) is not consistent with the figure legends. Helical parameters are not labeled parallelly in the right panel.
- 6) Line 139-140, is the 15 nm class with the best 2D average performance also the largest class? Otherwise, "most stable" is not an exact description due to lack of experimental evidence.
- 7) Line 143, outer layer extends to a 15 nm diameter across the center, not from the center.
- 8) Figure S2b and S2c, the negative-stain images for CARD8 FIINDUPA-CARD and FIINDUPA alone are not unified by the same scale bar.
- 9) As shown in figure 2a, NLRP1-CARD forms several thick or paired filaments, which does not happen to CARD8-CARD. Could the authors give some explanation?
- 10) Given the authors' description that CARD8-CARD filaments are more rigid and homogenous, the density quality for CARD8-CARD seems to be not as good as that for NLRP1-CARD even both filamentous structures are determined to the same resolution.
- 11) Figure 3a, diameter not radius is 8 nm.
- 12) The authors should provide the comparison of experimental and simulated power spectra to validate the quality of their helical model assignments.
- 13) Line 204, type II interfaces in NLRP1- and CARD8-CARD filaments show prominent electrostatic interactions, not electron density. Figure 2K left panel, "Arp459" is a typo.
- 14) Line 211, could the authors explain why non-interface mutation Q1388D in NLRP1-CARD could largely attenuate the activity to induce ASC specks?
- 15) Figure S5f, right panel, CARD8-CARD seems to induce oligomerization of ASC-CARD.
- 16) Line 335, figure 5a and 5b do not contain the data for FIINDUPA-CARD filament. Instead, figure S7a should be referred to.
- 17) Line 345-347, figure panels referred are incorrect and should be carefully checked. Expression of NLRP1-FL and CARD8-FIINDUPA-CARD in figure S7d and S7e cannot be observed by anti-mCherry western blotting.

Reviewer #2:

Remarks to the Author:

This paper describes an integrated structural, biochemical, and fluorescence imaging approach to analyze the activation mechanisms of human NLRP1 and CARD8, a process which is mediated by auto-proteolysis of the adjacent FIIND region into two sub-domains. The FIIND-UPA domain in turn appears to promote assembly of the adjacent, attached CARD into filaments at a lower concentration, and these filaments are decorated on their outside by a spiral of FIIND-UPA. These FIIND-UPA/CARD filaments can then interact in a rather stereo-specific manner with downstream effector CARDS of ASC or Caspase-1, respectively, utilizing a Type II interface located at the ends of the central CARD filaments.

The authors first determined a low resolution structure of the NLRP1-FIIND-UPA-CARD filament with negative stain EM; they have not tried to extend this work to higher resolution due in all likelihood to heterogeneity and flexibility issues. Even so, this structure clearly shows a central region with the diameter of a CARD helix decorated with a spiral of protein domains attributed to FIIND-UPA. They then determine four !! helical structures of NLRP1, CARD8, NLRC4 and ASC CARD filaments with nominal resolutions of 3.3 to 4.1 \AA . This represents a tremendous amount of work

and with perfect hindsight one can suggest that the choice of 300 kV imaging may not have been optimal for these rather thin and somewhat flexible filaments, as the impact of this tremendous effort would have been even greater if true side-chain resolution had been obtained. Even the NLRC4 CARD map as shown does not seem to show side chain density in the way that good maps at 3.3 Å resolution should. However, the density maps appear good enough to accurately dock the CARD helices with a reasonable register, as there is density to locate some of the larger side chains. Thus, in many of their interface comparisons they have a fairly accurate position for the amino acid in the helix, but its side chain position is only approximate. However, the authors have a wealth of structural information from other studies, backed up by fairly extensive mutation and biochemical studies to flesh out their hypothesis on how differences in one Type II interface can dictate the course of downstream activation. Importantly, they do not seem to over-interpret their data.

This paper represents a significant step forward in understanding how CARD based filaments can discriminate between possible downstream heterotypic CARD partners to co-assemble into the appropriate inflammasome complex.

A few comments in no particular order-

1. line 137: Figures 1 g,h described as cryo-EM, but it is in fact from negative stain EM.
2. It is important that the authors include a main figure diagram of one of the CARD filaments in some form, that shows where the Type I, II and III interfaces are, to orient readers. This could be best done on a thin ribbon model of the filament perhaps with zoomed in views for the interfaces. The map is not needed for this. This early clarification will make the rest of the story flow better.

I would also encourage the authors to use a sentence or two to explain what these interfaces are, why they have been identified etc. Just jumping into the CARD interfaces without some preamble or introduction will lose readers who may not be conversant with this somewhat finely honed terminology for the description of CARD surfaces.
3. line 235: Not sure why Tyr527, 531 are listed as contributing to a negatively charged surface in the CARD8-CARD filament, as these residues are expected to be neutral at physiological pH.
4. line 362: as stated earlier, the authors have also determined a cryo-EM structure of the helical NLRC4-CARD filament, with a stated resolution of 3.3Å; the density looks reasonable for docking the helices but at this resolution one would expect to see many side chains clearly resolved, which doesn't appear to be the case. Thus, the map seems to be more like 4Å resolution or a bit worse; is there a systematic over-estimation the actual resolution in these helical maps?
5. line 384: prionioid typo

Reviewer #3:

Remarks to the Author:

The manuscript by Gong et al. reports the structural study of NLRP1 and CARD8. By using refolded proteins, the author reconstituted filaments formed by FIINDUPA-CARD from NLRP1 or CARD8. Cryo-EM analysis showed that the filaments are a two-layered structure with the CARDS constituting the core and FIINDUPA the outer layer. They identified a unique Type II interface in the oligomerized NLRP1-CARD/CARD8-CARD/NLRC4-CARD that enable them to discriminate between ASC and pro-caspase-1 for assembly of distinct inflammasome complexes. This conclusion is further supported by cellular and biochemical data. The authors also provided evidence that FIINDUPA functions to facilitate the oligomerization of CARD. These results provide insight into human NLRP1 and

CARD8-inflammasome assembly. The structures of NLRP1-CARD and CARD8-CARD filaments are not new and they are very similar to those previously reported. The novelty of the current manuscript is that it revealed the structural basis of specificity for inflammasome signaling conferred by the conserved CARD domains, which is of high significance. The experiments were well performed, structural data are solid and conclusions are supported by the data presented. I recommend its publication in Nature Comm. I only have some minor questions about the manuscript (see below).

Many typos need correcting:

1. Page 10 line 280: "A remarkable level of specificity becomes apparent, as certain interactions are favored while others show patent incompatibility" should be "A remarkable level of specificity becomes apparent, as certain interactions are favored while others show potent incompatibility".
2. Page 10 line 282: "NLRP1-CARD and ASC-CARD is expected to be compatible, as the prominent HPH (a.a. 1452-1454) motif at the NLRP1 Type IIb interface can form ionic interactions with the ASC IIb residues AWN (a.a. 168-170)" should be "NLRP1-CARD and ASC-CARD is expected to be compatible, as the prominent HPH (a.a. 1452-1454) motif at the NLRP1 Type IIb interface can form ionic interactions with the ASC IIa residues AWN (a.a. 168-170)".
3. Page 10 line 286: "In other words, NLRP1-CARD uses the HPH motif on its Type IIa surface to preferentially engage ASC-CARD and disfavor CASP1-CARD" should be "In other words, NLRP1-CARD uses the HPH motif on its Type IIb surface to preferentially engage ASC-CARD and disfavor CASP1-CARD".
4. Line 445: In "Fig. 1. Human NLRP1 and CARD8 nucleate distinct inflammasome complexes", All the results in this figure are about NLRP1.
5. line 163: "CARD-FIINDUPA pre-oligomerizes" for "CARD8-FIINDUPA pre-oligomerizes"
6. line 196 : " (Fig.2d and 2i, Fig. S4c-d, Fig. S4a-d) in both filaments", It seems that the Fig. S4c-d and Fig. S4a-d could be merged into ONE (Fig.2d and 2i, Fig. S4a-d).
7. line 345: " Fig.S7c-f", No Fig.S7f in the figure.
8. in line 373: "(Dick et al., 2016)" format of the citation.
9. Page 6 line 153: "(Fig. 1i)" should be "(Fig. 1h)".
10. "Fig. 2e. ...shown in Fig. 1h-1i" should be "...shown in Fig. 1h".
11. LINE 529 Fig. 4f: "human CARD-CARD" for "human CARD8-CARD".
12. In Fig. S1d: "degredation" should be "degradation".
13. In Fig. S2b (line 790): CARD8-FINDUPA-CARD complexes should be CARD8-FIINDUPA-CARD complexes
14. In Fig. S2C (line 791): CARD8-FINDUPA ring-complexes should be CARD8-FIINDUPA ring-complexes
15. In Fig. S3d (line 799): NLRP1-CARDE1387R should be "NLRP1-CARDE1387R"
16. In Fig. S6 (line 823): "Full-length NLRP1- and CARD-mCherry filaments observed in Talabostat" should be "Full-length NLRP1- and CARD8-mCherry filaments observed in Talabostat".
17. In Fig. S7a (line 829-831): "Representative images of 293T-ASC-CARD- and CASP1-CARD-GFP responder cells transfected with mCherry-tagged NLRP1/CARD-FIINDUPA-CARD..." should be "Representative images of 293T-ASC-CARD-GFP and CASP1-CARD-GFP responder cells transfected with mCherry-tagged NLRP1/CARD8-FIINDUPA-CARD...".
in Fig. S7a image "Responder (mCherry)" should be "Responder (GFP)";
18. In Fig. S7a (line 833): "NLRP1-FINDUPA-CARD" should be "NLRP1-FIINDUPA-CARD".
19. In Fig. S7c (line 835): "CARD8-FINDUPA-CARD" should be "CARD8-FIINDUPA-CARD".
20. In Fig. S8c (line 847-848): "Percentage of filament/speck formation in 293T-ASC-CARD-GFP, ASC-GFP, CASP1-CARD-GFP and CASP1C285G-GFP responder cells transfected with NLRC4-CARD-mCherry" should be "Percentage of filament/speck formation in 293T-ASC-CARD-GFP, ASC-GFP, CASP1-CARD-GFP and CASP1C285G-GFP responder cells transfected with NLRC4-CARD-mCherry".
21. In Fig. S8d (line 851): "Confocal microscopy images of NLRC4-mCherry filaments..." should be "Confocal microscopy images of NLRC4-CARD-mCherry filaments...?"

REVIEWER COMMENTS

Reviewer #1 (Remarks to the Author):

In this work the authors investigate the structural basis of NLRP1- and CARD8-mediated inflammasome signalling using a combination of structural and cell-based approaches. They solved the filamentous structures of NLRP1-CARD and CARD8-CARD, and found that the FIINDUPA preceding the CARD is prone to oligomerization and triggers the functional FIINDUPA-CARD fragment of NLRP1 and CARD8 forms an ordered two-layer filament. They also investigated the potential structural mechanism by which NLRP1 and CARD8 could discriminate between ASC and pro-caspase-1, resulting distinct downstream signalling. This work has potential to be of broad interest in the innate immune field. However, there are still several points need to be addressed before the further consideration of their manuscript.

Major:

1. FIINDUPA-CARD of NLRP1 and CARD8, upon unleashed in response to activating signals, is actually the functional fragment which assembles into oligomeric inflammasome. As the authors showed, FIINDUPA is prone to self-oligomerize and likely bring the following CARD into proximity to form filamentous structure. This results in a regular two-layer filamentous assembly of FIINDUPA-CARD. The spiral assembly of FIINDUPA around the central CARD filament is supposed to be built on considerable homotypic interactions between neighboring FIINDUPA. The authors are encouraged to dissect the FIINDUPA interactions which are of great importance to uncover the precise molecular mechanism that initiates the inflammasome assembly of NLRP1 and CARD8.

We appreciate the reviewer's kind advice in further exploring the potential physiologically relevant UPA-UPA interaction interface. We followed the suggestions and conducted a series of mutagenesis in both bacterially expressed SUMO-NLRP1-UPA and mammalian NLRP1-FIIND^{UPA}-CARD based on homology modeling of published UPA structure of UNC5b (PDB:3D5B). Overall, our biochemical and functional data suggested a similar head-to-tail architecture of UPA ring wrapping around the CARD domain core filament, which is in agreement with two recently released preprint manuscripts (<https://doi.org/10.1101/2020.08.14.246132>; <https://doi.org/10.1101/2020.08.13.250241>). These data are shown in the new Figure 3, as shown below.

Fig.3 Gong et al

Fig. 3. Model of two-layer organization of NLRP1 and CARD8 FIIND^{UPA}-CARD filaments

- a. Top panel: SDS-PAGE NLRP1-FIIND^{UPA}-CARD mutants. Bottom panel: Native gel analysis of SUMO-NLRP1-FIIND^{UPA} constructs, stained by Krypton.
- b. ASC-GFP speck formation induced by wild-type NLRP1-FIIND^{UPA}-CARD and PP1278AG in 293T-ASC-GFP cells.
- c. Percentage of ASC-GFP specks induced by NLRP1-FIIND^{UPA}-CARD constructs in 293T-ASC-GFP cells. Cells were fixed 24 hours after transfection for fluorescence microscopy. P-value was calculated with One-way ANOVA, n=3 biological replicates.
- d. Side view and top view of modeled NLRP1-FIIND^{UPA}. Potential surface residues that may involve homotypic oligomerization were displayed using spheres, PP1278-1279, were highlighted in red color, R1240, RK1260, R12785 and F1320 are colored in orange.
- e. The top and side views of NLRP1-CARD filament and 6 modeled FIIND^{UPA} domains fitted into the FIIND^{UPA}-CARD filament density.
- f. A close-up view of the potential UPA-CARD linker orientation. A linker with 25 ~ 40 Å is likely to support a regular organization of the two-layer structure.
- g. The top and side views of CARD8-CARD filament and 6 modeled FIIND^{UPA} domains fitted into the CARD8-FIIND^{UPA}-CARD filament density.

These results, together with several other pieces of evidence presented in updated figures, also revealed a role of 9-22 amino-acid linker in organizing the two layer structure of NLRP1-FIIND^{UPA}-CARD filament.

2. There is no evidence to show that the ring-like oligomers formed by NLRP1 FIINDUPA alone recapitulate the organization and detailed interactions of FIINDUPA as in the spiral assembly of NLRP1 FIINDUPA-CARD filament. The heterogeneity of FIINDUPA oligomers only supports the intrinsic oligomerization property of FIINDUPA and cannot be simply applied to explain the spiral assembly of FIINDUPA in FIINDUPA-CARD filament. Thus, the observation that the oligomers formed by FIINDUPA alone and the outer layer of FIINDUPA-CARD filaments share a roughly similar size should not be conclusive.

The revised discussion has been rephrased to avoid too strong a claim on 'the ring-shape, 'formed ring-shaped or linear oligomers of heterogeneous stoichiometries *in vitro*.' We acknowledge that the SUMO tag and recombinant expression procedures may not fully recapitulate the morphology of the oligomers found *in vivo*. In the revised manuscript, we conclude that the oligomerization propensity is consistent with the two layer filament structure of the FIIND^{UPA}-CARD.

3. The authors should examine the topology of recombinant CARD8 FIINDUPA-CARD filaments at 3D-reconstruction level to demonstrate the similar two-layer filamentous assembly before they conclude a common model for both NLRP1 and CARD8 assembly.

We appreciate Reviewer 1's interest in further exploring the domain architecture of CARD8-FIINDUPA-CARD filament. We have endeavored during revision to study CARD8-FIIND^{UPA}-CARD in detail. Wild-type CARD8-FIIND^{UPA}-CARD is more heterogeneous than NLRP1, with a strong tendency to form short chunks instead of long filaments. The frequency of identifying such filament is so low that we could not obtain sufficient 2D information to build a preliminary

3D model as reviewer 1 suggested. through a systematic screening of recombinant construct, we realized that this was likely due to the short length (CARD8, 10 amino acid; NLRP1 19 amino acid) of the linker between CARD8-UPA domain and CARD domain. We therefore extended the linker length, and identified a 14 amino acid linker version of CARD8-FIIND^{UPA}-CARD construct that formed filaments at a higher frequency. Without applying a helical symmetry, we reconstituted a 3D density of CARD8-FIIND^{UPA}-CARD, and revealed a similar two layer-architecture of NLRP1-FIIND^{UPA}-CARD. After applying the same helical symmetry of NLRP1, the top view of this density resembled a circular shape that we observed from recombinant CARD8-FIIND^{UPA}-CARD. These results were summarized and presented in a new figure (Fig S2e-h, and S3c-i)

Fig. S2. Additional data for the analyzing the importance of the linker length in NLRP1 and CARD8 signaling

- a. NLRP1 constructs boundary and linker truncation analysis, using ASC speck formation as read out.
- b. Oligomerization, analyzed using native gel and probed by western blot, of selected NLRP1 constructs to validate the importance of flexible regions in NLRP1.
- c. Raw images of ASC speck formation induced by selected NLRP1 mutants.
- d. Sequence alignment of human, mouse and rat NLRP1, as well as human CARD8.
- e. CARD8 linker mutants (SNAP-FIIND^{UPA}-CARD) design.
- f. CARD8 linker mutants oligomerization analyzed by native gel.
- g. CARD8 linker mutants oligomerization analyzed by SDS-PAGE.
- h. CARD8 linker mutants analyzed by CASP1 speck formation.

Fig. S3. Additional data for the oligomerization of CARD8 constructs

- a. Size exclusion and SDS-PAGE profile of purified SUMO-CARD8-FIIND^{UPA}, Superdex S200 increase column.
 - b. Negative EM images of CARD8-FIND^{UPA} ring-complexes purified from *E. Coli*.
 - c. Negative EM images of CARD8-FIND^{UPA}-CARD complexes, wild type and the linker mutants.
 - d. Representative 2D class average images of CARD8-FIND^{UPA}-CARD.
 - e. 3D reconstruction of CARD8-FIND^{UPA}-CARD filament without imposing any symmetry.
 - f. Top view of reconstruction CARD8-FIND^{UPA}-CARD revealed a potential 11-12 unit per turn symmetry that is similar to NLRP1-FIND^{UPA}-CARD.
 - g. Negative stain EM images of CARD8-FIND^{UPA}-CARD when expressed without any fusion tags. A zoomed in view of a complex resembled the overall shape of the top view of helical reconstructed CARD8-FIND^{UPA}-CARD filament.
 - h. Top view of symmetry applied helical reconstruction of CARD8-FIND^{UPA}-CARD.
 - i. Side view of symmetry applied helical reconstruction of CARD8-FIND^{UPA}-CARD.
4. Experiment evidences are required to show that NLRP1-CARD and CARD8-CARD nucleate respectively ASC-CARD and CASP1-CARD filament assembly unidirectionally. This is the prerequisite that NLRP1-CARD and CARD8-CARD filaments utilize their top surfaces to prime the filamentous assembly of ASC-CARD and CASP1-CARD, respectively.

We attempted gold particle labelling experiments trying to capture the seeding events with only one end (NLRP1-CARD or CARD8-CARD) labelled with dark dots. However, the results were inconclusive, due to two reasons. Previous successful experiments, such as the CARMA complex, NLRP3-PYRIN or RIG-I/MAVS complex used seeds that were short chunks of filament segments. By contrast, NLRP1 and CARD8-CARD readily formed long filaments. The bodies of these 'seed' filaments became labeled with gold particles, preventing us from observing any growth of adaptor CARD filaments emanating from them in a directional manner. Secondly, using either anti-flag or anti-His gold particles alone could oligomerize the seeds and cause bundling of the filaments, against limiting our ability to observe unidirectional growth..

We then devised a new way demonstrating the unidirectional seeding of NLRP1 and CARD8. By adding a SNAP tag to ASC-/CASP1-CARD, these 'downstream' filaments appeared visibly thicker and could be easily distinguished from thinner, untagged NLRP1-CARD or CARD8-CARD filaments. By incubating SNAP tagged ASC-CARD and untagged NLRP1-CARD, as well as SNAP tagged CASP1-CARD and untagged CARD8-CARD, we observed only one end of thin NLRP1/CARD8 filament seeds were capable of stimulating filament formation of thick downstream ASC/CASP1 filaments. These results were summarized in a new figure (Fig, S9a-b). We highlighted the most obvious examples here.

Fig.S9 Gong et al

a

b

Fig. S9. Only one end of the NLRP1-CARD and CARD-CARD filament is capable of seeding ASC-CARD or CASP1-CARD

- a. Negative stain EM images of ASC-CARD tagged with non-cleavage SNAP tags seeded by naked NLRP1-CARD filament segments.
- b. Negative stain EM images of CASP1-CARD tagged with non-cleavage SNAP tags seeded by naked CARD8-CARD filament segments.

5. The complementarity of surface charge distribution is lack of precision to well explain why NLRP1-CARD and CARD8-CARD filaments nucleate different downstream CARD. Based on the surface charge distribution shown by the authors, ASC-CARD filament would not prefer to engage CASP1-CARD, which is evidently not the fact. Considering that all the structurally characterized CARD filaments share highly similar helical symmetry parameters, it is likely that the priming CARD filament would elongate by incorporating downstream CARD via the same organization mode. Thus, inspecting the compatibility of all the three types of interface between the primer and downstream CARD is necessary.

The original Figures might not be clear enough to demonstrate the surface complementarity we would like to convey to Reviewer 1. We have reworked the images to highlight the important areas and the particular Type II interface density bridges. ASC-CARD top surface (composed of Type Ib and IIb interfaces) does indeed have a good match with the bottom surface of CASP1 (composed of Type Ia and IIa interfaces). The new figure (Now Fig 4c-h) is shown below.

Fig. 4. Oligomeric interfaces on inflammasome CARD domains

- a. Left panel: side-view of electrostatic potential distribution of a segment of NLRP1-CARD filament (red, -10 C/m^2 ; blue, 10 C/m^2 ; generated in Chimera). Right panel: electrostatic potential map of a NLRP1-CARD monomer in the same orientation as the filamentous oligomer.
- b. Left panel: side-view of electrostatic potential distribution of a segment of CARD8-CARD filament (red, -10 C/m^2 ; blue, 10 C/m^2). Right panel: electrostatic potential map of a CARD8-CARD monomer in the same orientation.
- c. Atomic model cartoon and the directly observed density map of a NLRP1-CARD dimer within the filamentous complex, centered at the Type II junction interface. The four residues, His1452-His1454 on the Type IIb interface and the Ser1435-Asp1437 on the Type IIa interface were highlighted and shown in the 'stick' mode.
- d. Atomic model cartoon and the directly observed density map of CARD8-CARD dimer in the filament, centered at the Type II junction interface. The four residues, Asp525-Tyr527 on the Type IIb interface and the Lys508-Leu512 on the Type IIa interface were highlighted and shown in the 'stick' mode.
- e. Electrostatic potential map of the top view (Type IIb interface, highlighted in yellow dashed area) of the NLRP1-CARD filament.
- f. Electrostatic potential map of the top view (Type IIb interface, highlighted in yellow dashed area) of CARD8-CARD filament.
- g. Electrostatic potential map of the bottom (Type IIa highlighted in yellow dashed area) and top (Type IIb highlighted in yellow dashed area) views of the NLRP1-CARD filament.
- h. Electrostatic potential map of the bottom (Type IIa highlighted in yellow dashed area) and top (Type IIb highlighted in yellow dashed area) views of the CARD8-CARD filament.

Furthermore, as shown in the cartoon illustrations of Fig 4e-f, the bottom-up filament seeding interactions mainly involve only Type I and Type II interfaces. In the particular cases of NLRP1 and CARD8 CARD, Type I interactions are fairly weak, and Type II interactions are the dominant forces stabilizing the stacking of CARD domains within the filament. This is supported by the fact that the Type II junctions (Fig 4c-d) are the strongest density linking different CARD monomers together in NLRP1 and CARD8 CARD domain filaments. This is drastically different from RIG-I/MDA5/MAVS, in which Type I interactions dominate.

6. The highlighted dominant role of type II interface in discriminating downstream CARD by NLRP1-CARD and CARD8-CARD should be proved by mutagenesis analyses, including loss of preformed intermolecular interactions such as NLRP1-ASC or CARD8-CASP1, and gain of nonexistent intermolecular interactions such as NLRP1-CASP1 or CARD8-ASC.

We agree with Reviewer 1 that the best way to demonstrate the signaling specificity would be to rewire the outcome through mutagenesis. We attempted mutating the top surfaces (Type IIa and IIb interfaces) of NLRP1-CARD and CARD8-CARD extensively. However, all said mutants NLRP1-CARD and CARD8-CARD filament formation, suggesting that these residues are critical for self-oligomerization of NLRP1 and CARD8, which is a prerequisite for NLRP1 or CARD8 to engage downstream partners. Thus, it may not be possible to design gain function mutants in NLRP1 and CARD8 that can still form filaments by themselves. Instead, we would like to bring Reviewer 1's attention to Fig 2l-m, here, our filament structures explain the

biochemical and cellular behaviors of almost 30 surface charge mutations, in NLRP1 and CARD8 CARD domains. For instance, In particular, Q1434R mutation in NLRP1 has elevated signaling due to improved surface complementarity after the mutation.

Minor:

1) Introduction, line 65, CARD8 is conceptually not a NLR protein, but a FIIND domain-containing protein closely related to NLRP1 which can activate ASC-independent inflammasome.

Text updated to 'NLRP1 and a related receptor'.

2) Considering that the filamentous structure of PYRIN or CARD domain was well studied before and likely a universal mechanism for inflammasome assembly, this documented background should be referred to in the introduction part.

Additional description and citations were included 'Inflammasome sensor NLRs can be grouped into two categories based on their DDs: PYRIN domain (PYD) and caspase activation and recruitment domain (CARD), both domains were found to form filamentous complexes to initiate the inflammasome formation'.

3) Figure 1e, the images of negative-stain EM are of less quality than required. In the field shown, 100 uM of NLRP1 FIINDUPA-CARD does not mainly form long filament by self-assembly; also the low-order oligomerization of FIINDUPA-CARD at 10 uM could not be distinguished from the supposed monomeric CARD at 100 uM. The author should explain how they estimate the size of the oligomers.

The estimation of the oligomer size is based on native gel analysis, slower migrating population (oligomer > 1 MDa, based on migration speed, Fig 1d) of NLRP1-FIIND^{UPA}-CARD is visible at much lower concentration.

4) Line 131-132, comparison of the oligomers formed by diverse NLRP1 fragments would be better demonstrated by back-to-back negative-stain images for all the fragments (with the same scale bar). In the size-exclusion profiles such as figure S1c and S1d, the molecular weight standard should be indicated.

Fig S1 was updated accordingly to address the comment. One additional panel Fig S1e, was included to directly compare NLRP1-FIIND^{UPA}-CARD, NLRP1-CARD and Sumo-NLRP1-FIIND^{UPA} constructs, under the same magnification.

5) The authors should clarify whether figure 1g is the representative 2D classification for cryo-EM or negative-stain EM raw images. The description in the text (line 137) is not consistent with the figure legends. Helical parameters are not labeled parallelly in the right panel.

We thank Reviewer 1 for pointing this out. We have updated the text to improve clarity. These were negative stain images. Cryo-EM experiments were not successful due to low sample density.

6) Line 139-140, is the 15 nm class with the best 2D average performance also the largest class? Otherwise, "most stable" is not an exact description due to lack of experimental evidence.

Text was updated to 'most abundant', to improve clarity.

7) Line 143, outer layer extends to a 15 nm diameter across the center, not from the center.

We thank Reviewer 1 for pointing this out. Text was updated accordingly.

8) Figure S2b and S2c, the negative-stain images for CARD8 FIINDUPA-CARD and FIINDUPA alone are not unified by the same scale bar.

This figure was extensively updated (now Fig S3b-c), and the scale bar was individually calibrated.

9) As shown in figure 2a, NLRP1-CARD forms several thick or paired filaments, which does not happen to CARD8-CARD. Could the authors give some explanation?

The side surface of CARD8-CARD filament is largely negatively charged (Fig 4b), making self-bundling less favorable. In addition, the last helix of CARD8-CARD is more rigid and pointing outward, creating additional hindrance for potential side-to-side interactions. By contrast, the side surfaces of NLRP1-CARD are more favorable for potential self-bundling, and the last 8 amino acids of NLRP1-CARD domain promote bundling. As a result, the best NLRP1-CARD dataset we collected was using a 1374-1465 truncated construct.

10) Given the authors' description that CARD8-CARD filaments are more rigid and homogenous, the density quality for CARD8-CARD seems to be not as good as that for NLRP1-CARD even both filamentous structures are determined to the same resolution.

Reviewer 1 made an interesting observation on the correlation of filament morphology and density qualities.

In our opinion, the filament morphology is more linked to the heterogeneity of the helical parameters. NLRP1-CARD filament may have several possible favorable helical parameters 'co-existing' in its filaments. In our manuscript, we focused on analyzing one predominant type of NLRP1-CARD helical arrangement, after 3D classification. While in the case of CARD8-CARD, there is probably only one.

The quality of the density is more related to how well the amino acids within the monomer are refined. When analyzing the densities, we conduct minimal density polishing, applying only gentle global B factor based sharpening. Thus, in general, our densities look a bit 'bulky'. In the particular case of CARD8-CARD filament density, lacking sufficient Type I interactions may render individual CARD8 CARD domains having intrinsic higher B factors.

So our interpretation to the observation is, CARD8-CARD filament has uniform helical parameters, but individual CARD domain may tumble within its own 'sphere'.

11) Figure 3a, diameter not radius is 8 nm.

Corrected.

12) The authors should provide the comparison of experimental and simulated power spectra to validate the quality of their helical model assignments.

Power spectrum comparison of NLRP1-CARD, CARD8-CARD, ASC-CARD, NLRC4-CARD and NLRP1-FIIND^{UPA}-CARD, were included as additional supporting data of the manuscript, and attached below. Note that CARD8-FIIND^{UPA}-CARD's power spectrum of the 2D average is too weak to be analyzed directly, we performed a non-symmetry reconstruction for CARD8-FIIND^{UPA}-CARD, and demonstrated its two layer architecture and the 11-12 unit per turn outer layer feature, without implying any helical symmetry.

NLRP1-CARD filament Power Spectrum
 Theoretical (simulated 1 Å, top Vs. real 2D class image, bottom)

CARD8-CARD filament Power Spectrum
 Theoretical (simulated 1 Å, top Vs. real 2D class image, bottom)

ASC-CARD filament Power Spectrum
 Theoretical (simulated 1 Å, top Vs. real 2D class image, bottom)

NLRC4-CARD filament Power Spectrum
 Theoretical (simulated 1 Å, top Vs. real 2D class image, bottom)

NLRP1-FIIND^{UPA}-CARD filament Power Spectrum
 Theoretical (simulated 20 Å, top Vs. real 2D class image, bottom)

13) Line 204, type II interfaces in NLRP1- and CARD8-CARD filaments show prominent electrostatic interactions, not electron density. Figure 2K left panel, “Arp459” is a typo.

Corrected.

14) Line 211, could the authors explain why non-interface mutation Q1388D in NLRP1-CARD could largely attenuate the activity to induce ASC specks?

This observation also puzzled us. We designed the mutations and conducted the cellular assays in a double blind method, and the expression level is monitored via its own mCherry tags. That being said, it is not uncommon to have loss-of-function mutants from time to time, even if they do not appear to directly impact the key structural features. Potential explanations could be, (1) Q1388 is site of a important post-translational mutation; (2) Q1388D affects the conformational change when FIIND^{UPA}-CARD transit from monomer to oligomer; (3) Q1388 is important for interaction with a unknown interaction partner, which influences the life cycle or cellular localization of NLRP1.

15) Figure S5f, right panel, CARD8-CARD seems to induce oligomerization of ASC-CARD.

In recombinant assays, CARD8-CARD does activate ASC-CARD oligomerization directl, but is at least 10 times weaker than NLRP1-CARDStructurally speaking, CARD8-ASC interaction is unfavorable, not impossible. Thus, we chose to describe it as ‘CARD8 preferentially stimulated CASP1-CARD oligomerization.’

16) Line 335, figure 5a and 5b do not contain the data for FIINDUPA-CARD filament. Instead, figure S7a should be referred to.

Thanks for pointing this out. We have corrected the figure citation here.

17) Line 345-347, figure panels referred are incorrect and should be carefully checked. Expression of NLRP1-FL and CARD8-FIINDUPA-CARD in figure S7d and S7e cannot be observed by anti-mCherry western blotting.

Full length NLRP1 and CARD8-FIIND^{UPA}-CARD were expressed at the lowest levels among all the constructs texted in this panel. Yet they were the most potent at inducing pyroptosis (along with NLRP1-FIIND^{UPA}-CARD). Hence, in Cas9 control cells transfected with full length NLRP1 and CARD8-FIIND^{UPA}-CARD, all the transfected cells had died or were dying of pyroptosis at the time of harvesting. This led to low/minimal levels of transfected proteins detected by Western blot. In support of this, the expression of the pyroptosis inducing constructs (e.g. full length NLRP1, NLRP1-FIIND^{UPA}-CARD and CARD8-FIIND^{UPA}-CARD) all appeared higher in ASC KO or CASP1 KO cells that could not undergo pyroptosis (e.g. lane 2 vs lane 6 in d; lane 2 vs lane 8 in e)

#Additional comments on a recently published manuscript from Hao Wu's group, also reporting the structures of filamentous complexes of NLRP1 and CARD8 CARD domains.

Comparing the three structures, the double layer NLRP1-CARD density reported by Hao Wu's group, our single layer NLRP1-CARD alone, and our low resolution NLRP1-UPA-CARD, we think there is actually strong synergies that may explain the different aspects of activated NLRP1 signalosome.

First of all, both of our studies identified the same helical parameter, diameter, and orientations of monomer for the inner NLRP1-CARD filament

Secondly, at low resolution, it was very challenging to directly confirm the identity of the outer layer density. We stand by our claim that the outer layer density consists of UPA due to the following observations:

1. Comparing the two layer NLRP1-CARD structure from Hao Wu's group with our model, we noticed that the outer layer helical parameters are different in both rotational angles and lateral translational distances. In our case, we noticed arrangements of 10-12 density blobs per turn on the outside layer (4.5-5.5 unit per turn judging from 2D average image directly), compared to 7 unit per turn (identical helical parameter to the inner layer) on the outside in the density from the Wu lab. So our guess is that when we prepared these filaments differently, using slightly different constructs with different boundaries (particularly C-terminal residues of NLRP1-CARD, since we truncated the last 8 amino acids to discourage the bundling of NLRP1-CARD filaments) and tags, we may encourage the filament formation of one type or the other. Despite low at resolution, our UPA-CARD density has an outer layer with distinct helical organization.
2. SDS-PAGE showed that UPA-CARD were intact in our two layer filament.
3. We did boundary/linker optimization to avoid excessive interactions between NLRP1 CARD-CARD, and in this way obtained 3.7A thin filament of NLRP1-CARD alone. After optimization, NLRP1-CARD alone filament could be obtained without refolding.
4. After fitting a UPA domain homologous model (3G5B) into the outer layer blobs of our low resolution UPA-CARD density, we observed reasonable lateral interactions/distances between UPA domains.
5. NLRP1-UPA domains alone formed ring-like or arc like structures in vitro

6. Hao Wu's study directly applied the helical symmetry to the outer layer, which would override any other alternative helical symmetry information, thus making resolving potential UPA domain density more challenging.

7. Our two-layer model offers a direct explanation as to why UPA-UPA interaction is required for NLRP1 activation. These results are confirmed by two additional pre-print publications from our group and Hao Wu's group.

That being said, both of our studies could be correct. The structure from the Hao Wu lab demonstrated a very real possibility that the CARD domain may fit/inserted into the outer layer of the actual NLRP1 signalosome in cells. Building upon our analogy of a corn cob, the *in vivo* scenario for NLRP1 inflammasome activation might resemble the picture below, where the stacking of the NLRP1-FIIND^{UPA}-CARD subunits may not be as homogeneous and the insertion of an additional 'grain' can very well occur without grossly altering the overall complex structure or the signaling outcome. In fact, even in our protocol, the NLRP1-FIIND^{UPA}-CARD double layer filaments were flexible and we always saw a mix of helical organizations in the outer layer, potentially due to occasional and random insertion of CARD domain into the outer layer, as Hao Wu's structure suggested.

Reviewer #2 (Remarks to the Author):

This paper describes an integrated structural, biochemical, and fluorescence imaging approach to analyze the activation mechanisms of human NLRP1 and CARD8, a process which is mediated by auto-proteolysis of the adjacent FIIND region into two sub-domains. The FIIND-UPA domain in turn appears to promote assembly of the adjacent, attached CARD into filaments at a lower concentration, and these filaments are decorated on their outside by a spiral of FIIND-UPA. These FIIND-UPA/CARD filaments can then interact in a rather stereo-specific

manner with downstream effector CARDS of ASC or Caspase-1, respectively, utilizing a Type II interface located at the ends of the central CARD filaments.

The authors first determined a low resolution structure of the NLRP1-FIIND-UPA-CARD filament with negative stain EM; they have not tried to extend this work to higher resolution due in all likelihood to heterogeneity and flexibility issues. Even so, this structure clearly shows a central region with the diameter of a CARD helix decorated with a spiral of protein domains attributed to FIIND-UPA. They then determine four helical structures of NLRP1, CARD8, NLRC4 and ASC CARD filaments with nominal resolutions of 3.3 to 4.1 Å. This represents a tremendous amount of work and with perfect hindsight one can suggest that the choice of 300 kV imaging may not have been optimal for these rather thin and somewhat flexible filaments, as the impact of this tremendous effort would have been even greater if true side-chain resolution had been obtained. Even the NLRC4 CARD map as shown does not seem to show side chain density in the way that good maps at 3.3 Å resolution should. However, the density maps appear good enough to accurately dock the CARD helices with a reasonable register, as there is density to locate some of the larger side chains. Thus, in many of their interface comparisons they have a fairly accurate position for the amino acid in the helix, but its side chain position is only approximate. However, the authors have a wealth of structural information from other studies, backed up by fairly extensive mutation and biochemical studies to flesh out their hypothesis on how differences in one Type II interface can dictate the course of downstream activation. Importantly, they do not seem to over-interpret their data.

This paper represents a significant step forward in understanding how CARD based filaments can discriminate between possible downstream heterotypic CARD partners to co-assemble into the appropriate inflammasome complex.

We appreciate Reviewer 2's recognition of our work.

A few comments in no particular order-

1. line 137: Figures 1 g,h described as cryo-EM, but it is in fact from negative stain EM.

We have corrected this error.

2. It is important that the authors include a main figure diagram of one of the CARD filaments in some form, that shows where the Type I, II and III interfaces are, to orient readers. This could be best done on a thin ribbon model of the filament perhaps with zoomed in views for the interfaces. The map is not needed for this. This early clarification will make the rest of the story flow better.

I would also encourage the authors to use a sentence or two to explain what these interfaces are, why they have been identified etc. Just jumping into the CARD interfaces without some preamble or introduction will lose readers who may not be conversant with this somewhat finely honed terminology for the description of CARD surfaces.

We thank Reviewer 2 for the kind suggestions. We have updated the figure to improve the clarity, attached below are the update panels of Fig 2. Additional text was included to explain the naming tradition.

The definition of the Type I-II interface follows the naming convention first used in describing the MyDDosome complex, and subsequently widely adopted in describing Death Domain filaments, such as MAVS-CARD. When positioning the first alpha helix of CARD domain downward, and the sixth alpha helix upward, the monomer on top is called 'a' monomer, and all three neighboring monomers are called 'b' monomers. From left to right, each 'b' monomers have a unique interaction interface with 'a' monomer, and from left to right, named as Type I, Type II and Type III interfaces, respectively. The corresponding surfaces on 'a' monomer are Type Ia, IIa and IIIa surfaces, and the surfaces on the three 'b' monomers are Type Ib, IIb and IIIb surfaces. All surface interactions within Death Domain filaments could be grouped into one of these types of interfaces (Fig. 2d and 2i). As compared to other DD complexes, the Type II interfaces in NLRP1- and CARD8-CARD filaments display stronger and more prominent electrostatic interactions than Type I and Type III (Fig. 2j-k). This is particularly notable in CARD8-CARD filaments, where Type Ia and Ib surfaces are separated from each other (Fig. 2j-k).

3. line 235: Not sure why Tyr527, 531 are listed as contributing to a negatively charged surface in the CARD8-CARD filament, as these residues are expected to be neutral at physiological pH.

We thank Reviewer 2 for pointing this out. We have updated the description to 'partially negatively charged patch', to describe Tyr527, 531 and Asp525.

4. line 362: as stated earlier, the authors have also determined a cryo-EM structure of the helical NLRC4-CARD filament, with a stated resolution of 3.3Å; the density looks reasonable for docking the helices but at this resolution one would expect to see many side chains clearly resolved, which doesn't appear to be the case. Thus, the map seems to be more like 4Å

resolution or a bit worse; is there a systematic over-estimation the actual resolution in these helical maps?

We maintain a relatively conservative stance in presenting polished cryo-EM densities. If we did more aggressive polishing on our datasets, in particular the NLRC4-CARD 3.3Å one, we could obtain sharp density features, shown here is the 3.3Å density after a -150 B factor polishing using CisTEM, displayed in ChimeraX. However, we realized that the model building statistics were not improved when using 'sharp' maps, hence we decided to present our side chain allocation using less polished maps. In terms of resolution calculation, we followed strict gold standard FSC calculation, and did this using both RELION 3Drefine/postprocess modules, and the EMDB server, so we have confidence in our conclusions.

5. line 384: prionioid typo

Thanks for pointing this out. Corrected.

Reviewer #3 (Remarks to the Author):

The manuscript by Gong et al. reports the structural study of NLRP1 and CARD8. By using refolded proteins, the author reconstituted filaments formed by FIINDUPA-CARD from NLRP1 or CARD8. Cryo-EM analysis showed that the filaments are a two-layered structure with the CARDS constituting the core and FIINDUPA the outer layer. They identified a unique Type II

interface in the oligomerized NLRP1-CARD/CARD8-CARD/NLRC4-CARD that enable them to discriminate between ASC and pro-caspase-1 for assembly of distinct inflammasome complexes. This conclusion is further supported by cellular and biochemical data. The authors also provided evidence that FIINDUPA functions to facilitate the oligomerization of CARD. These results provide insight into human NLRP1 and CARD8-inflammasome assembly. The structures of NLRP1-CARD and CARD8-CARD filaments are not new and they are very similar to those previously reported. The novelty of the current manuscript is that it revealed the structural basis of specificity for inflammasome signaling conferred by the conserved CARD domains, which is of high significance. The experiments were well performed, structural data are solid and conclusions are supported by the data presented. I recommend its publication in Nature Comm. I only have some minor questions about the manuscript (see below).

We thank Reviewer 3's critique and detailed suggestions.

Many typos need correcting:

1. Page 10 line 280: "A remarkable level of specificity becomes apparent, as certain interactions are favored while others show patent incompatibility" should be "A remarkable level of specificity becomes apparent, as certain interactions are favored while others show potent incompatibility".

Reviewer 3 is correct. Thanks for pointing out this typo. Corrected.

2. Page 10 line 282: "NLRP1-CARD and ASC-CARD is expected to be compatible, as the prominent HPH (a.a. 1452-1454) motif at the NLRP1 Type IIb interface can form ionic interactions with the ASC IIb residues AWN (a.a. 168-170)" should be "NLRP1-CARD and ASC-CARD is expected to be compatible, as the prominent HPH (a.a. 1452-1454) motif at the NLRP1 Type IIb interface can form ionic interactions with the ASC IIa residues AWN (a.a. 168-170)".

Thanks for pointing out. Corrected.

3. Page 10 line 286: "In other words, NLRP1-CARD uses the HPH motif on its Type IIa surface to preferentially engage ASC-CARD and disfavor CASP1-CARD" should be "In other words, NLRP1-CARD uses the HPH motif on its Type IIb surface to preferentially engage ASC-CARD and disfavor CASP1-CARD".

Corrected.

4. Line 445: In "Fig. 1. Human NLRP1 and CARD8 nucleate distinct inflammasome complexes", All the results in this figure are about NLRP1.

Updated to 'Human NLRP1 inflammasome complexes.'

5. line 163: "CARD-FIINDUPA pre-oligomerizes" for "CARD8-FIINDUPA pre-oligomerizes"

Corrected.

6. line 196 : “ (Fig.2d and 2i, Fig. S4c-d, Fig. S4a-d) in both filaments”, It seems that the Fig. S4c-d and Fig. S4a-d could be merged into ONE (Fig.2d and 2i, Fig. S4a-d).

Reviewer 3 is correct. We have updated the citation.

7. line 345: “ Fig.S7c-f”, No Fig.S7f in the figure.

Updated.

8. in line 373: “(Dick et al., 2016)” format of the citation.

Citation relinked. We thank Reviewer 3 for pointing this out.

9. Page 6 line 153: “(Fig. 1i)” should be “(Fig. 1h)”.

Updated.

10. “Fig. 2e. ...shown in Fig. 1h-1i” should be “...shown in Fig. 1h”.

Updated.

11. LINE 529 Fig. 4f: “human CARD-CARD” for “human CARD8-CARD”.

Updated.

12. In Fig. S1d: “degredation” should be “degradation”.

Updated.

13. In Fig. S2b (line 790): CARD8-FINDUPA-CARD complexes should be CARD8-FIINDUPA-CARD complexes

Updated.

14. In Fig. S2C (line 791): CARD8-FINDUPA ring-complexes should be CARD8-FIINDUPA ring-complexes

updated.

15. In Fig. S3d (line 799): NLRP1-CARDE1387R should be “NLRP1-CARDE1387R”

updated.

16. In Fig. S6 (line 823): “Full-length NLRP1- and CARD-mCherry filaments observed in Talabostat” should be “Full-length NLRP1- and CARD8-mCherry filaments observed in Talabostat”.

Updated.

17. In Fig. S7a (line 829-831): “Representative images of 293T-ASC-CARD- and CASP1-CARD-GFP responder cells transfected with mCherry-tagged NLRP1/CARD-FIINDUPA-CARD...” should be “Representative images of 293T-ASC-CARD-GFP and CASP1-CARD-GFP responder cells transfected with mCherry-tagged NLRP1/CARD8-FIINDUPA-CARD...”.
in Fig. S7a image “Responder (mCherry)” should be “Responder (GFP)”;

Updated.

18. In Fig. S7a (line 833): “NLRP1-FINDUPA-CARD” should be “NLRP1-FIINDUPA-CARD”.

Updated.

19. In Fig. S7c (line 835): “CARD8-FINDUPA-CARD” should be “CARD8-FIINDUPA-CARD”.

Updated.

20. In Fig. S8c (line 847-848): “Percentage of filament/speck formation in 293T-ASC-CARD-GFP, ASC-GFP, CASP1-CARD-GFP and CASP1C285G-GFP responder cells transfected with NLRC4-CARD-mCherry” should be “Percentage of filament/speck formation in 293T-ASC-CARD-GFP, ASC-GFP, CASP1-CARD-GFP and CASP1C285G-GFP responder cells transfected with NLRC4-CARD-mCherry”.

Thanks for pointing this out. Updated.

21. In Fig. S8d (line 851): “Confocal microscopy images of NLRC4-mCherry filaments...” should be “Confocal microscopy images of NLRC4-CARD-mCherry filaments...”?

Updated.

Reviewers' Comments:

Reviewer #1:

Remarks to the Author:

The authors made great efforts to do necessary experiments and revise the writing and the graphical presentation to make their findings clear and rigorous. The revised manuscript is much improved. Most raised questions have been well addressed. I would like to support to publish the revised paper after they address few typos and one point that should be discussed.

Major point:

The authors proposed a “head-to-tail” mode for FIINDUPA self-assembly in the active two-layer filamentous assembly of FIINDUPA-CARD. However, homotypic interactions of two neighboring FIINDUPA were determined to adopt a “head-to-head” architecture in DPP9-bound autoinhibited NLRP1 structure as shown by the referred two preprint manuscripts (<https://doi.org/10.1101/2020.08.14.246132>; <https://doi.org/10.1101/2020.08.13.250241>). The authors should discuss this discrepancy.

Typos:

Page 7, the second line from the bottom, CARD-FIINDUPA-CARD should be CARD8-FIINDUPA-CARD.

Page 8, line 8. proximityto.

Page 9, line 17, Type I-II interface should be Type I-III interface.

Page 17, line 16, nextolved.

Reply to additional comment

Reviewer 1's comment

The authors proposed a “head-to-tail” mode for FIINDUPA self-assembly in the active two-layer filamentous assembly of FIINDUPA-CARD. However, homotypic interactions of two neighboring FIINDUPA were determined to adopt a “head-to-head” architecture in DPP9-bound autoinhibited NLRP1 structure as shown by the referred two preprint manuscripts (<https://doi.org/10.1101/2020.08.14.246132>; <https://doi.org/10.1101/2020.08.13.250241>). The authors should discuss this discrepancy.

We appreciate Reviewer 1's acknowledgement of our effort in addressing previous comments. He/she have read all the recent literature carefully and identified a discrepancy in how we and other groups describe the UPA domain oligomerization. As shown above, after analyzing our proposed model and the published UPA dimer structures in the preprints (<https://doi.org/10.1101/2020.08.14.246132>; <https://doi.org/10.1101/2020.08.13.250241>), we realize the discrepancy is purely due to 'angle of description'. We are essentially describing the same way of organization. In the preprints, viewing from top, the 'head' surfaces are side by side. Viewing from the same angle, our predicted model (Figure 3d) is also 'head to head'. When we referred to 'head to tail', it was mainly describing when viewing from the side (Figure 3d), the 'head' surface of UPA is touching the 'tail' surface of its neighboring UPA, so that a continuous UPA oligomer is possible. From the same side view, the UPA oligomerization is also considered 'head to tail'. We have attached a slide here to facilitate the explanation. In brief, our model agrees with the preprints very well. To improve the clarity, we have modified the Figure 3d accordingly, and update the main text description to better reflect the structural comparison. In particular, change the 'head-to-tail' to 'side-by-side', to avoid unnecessary confusion.

‘Taken together, these findings suggest that NLRP1-FIIND^{UPA} employs one large hydrophobic surface patch centred around a.a. 1278-1279 (red) to self-oligomerize in a ‘side-by-side’ arrangement (Fig. 3d), giving rise to the outer ring (Fig. 3e) that wraps around the CARD core in the FIIND^{UPA}-CARD filaments (Fig. 1j). This organization closely resembles the FIIND^{UPA}-dimer structures seen in two recently reported NLRP1-DPP9 complexes ^{19,21}.’

In this way, we hope our manuscript could resolve any concerns from the audience. Here attached the updated Figure 3d.

Updated Figure 3 legend.

C. Percentage of ASC-GFP specks induced by NLRP1-FIIND^{UPA}-CARD constructs in 293T-ASC-GFP cells. Cells were fixed 24 hours after transfection for fluorescence microscopy. P-value was calculated with One-way ANOVA, n=3 biological replicates.

D. Side view and top view of FIIND^{UPA}-dimer structures reported by other groups, and our modeled oligomer interface of NLRP1-FIIND^{UPA}. Potential surface residues that may involve homotypic oligomerization were displayed using spheres, PP1278-1279, were highlighted in red color, R1240, RK1260, R12785 and F1320 are colored in orange.